behaviour/ecology/evolution

behaviour, non-invasive, physiology, inflammation, field studies

**Author for correspondence:**
James P. Higham
e-mail: jhigham@nyu.edu

# Urinary suPAR: a non-invasive biomarker of infection and tissue inflammation for use in studies of large free-ranging mammals

James P. Higham[1], Christiane Stahl-Hennig[2] and Michael Heistermann[3]

[1]Department of Anthropology, New York University, 25 Waverly Place, New York, NY 10003, USA
[2]Unit of Infection Models, and [3]Endocrinology Laboratory, German Primate Center, Leibniz Institute for Primate Research, Kellnerweg 4, Göttingen 37077, Germany

JPH, 0000-0002-1133-2030

Studies of large free-ranging mammals incorporating physiological measurements typically require the collection of urine or faecal samples, due to ethical and practical concerns over trapping or darting animals. However, there is a dearth of validated biomarkers of immune activation and inflammation that can be measured non-invasively. We here evaluate the utility of urinary measurements of the soluble form of the urokinase plasminogen activator receptor (suPAR), for use as a health marker in studies of wild large mammals. We investigate how urinary suPAR concentrations change in response to viral infection and surgical trauma (inflammation), comparing it to the measurement of a marker of cellular immune activation, urinary neopterin (uNEO), in captive rhesus macaques. We then test the field utility of urinary suPAR, assessing the effects of soil and faecal contamination, sunlight, storage at different temperatures, freeze–thaw cycles, and lyophilization. We find that suPAR concentrations rise markedly in response to both infection and surgery-associated inflammation, unlike uNEO concentrations, which only rise in response to the former. Our field validation demonstrates that urinary suPAR is reasonably robust to many of the issues associated with field collection, sample processing, and storage, as long as samples can be stored in a freezer. Urinary suPAR is thus a promising biomarker applicable for monitoring various aspects of health in wild primates and potentially also other large mammals.

# 1. Introduction

Studies of the ecology and evolution of wild vertebrates benefit greatly from being able to undertake measurements of individual health and condition [1–3]. For smaller-bodied taxa, animals can be trapped (rodents) or netted (birds, bats). This allows direct measurements of body condition to be taken (e.g. body mass, mouse-tailed bats [4]), or blood samples collected for analyses of hormones (e.g. corticosterone in Belding's ground squirrels [5]; dark-eyed juncos [6]) or biomarkers of immune activation (e.g. barn swallows [7]) and inflammation (e.g. spectacled thrushes [8]). Such measurements are critical for the monitoring of individuals for studies of population health and conservation [9,10], but also for a range of ecological and evolutionary questions [1,11]. These include topics such as disease ecology and the impacts of the physical and social environment on health and disease [12–14], the evolution of immunity and its role in species diversification [15], and the evolution of the MHC region and its role in mate choice [16].

Studies of large-bodied wild and free-ranging mammals present specific problems due to the physical difficulties, expense, and ethics associated with capturing and anaesthetizing them for blood collection. Nonetheless, in studies of large mammals such as buffalos [17,18], black rhinos [19] and some species of primates [20,21], sometimes such trapping has been undertaken, especially historically. Given the risks to the animals involved, wherever possible, wildlife health monitoring should focus on different aspects of an animal's physiology that can be measured non-invasively, particularly when dealing with endangered species. In some studies, non-invasive visual assessments of health or condition, such as the visual estimation of body fat [22], and signs of obvious wounds [13], or infections [23] have been conducted. Non-invasive measurements from urine and faecal samples are also commonplace in these taxa, such as for the quantification of faecal parasite load [24,25], and for hormone analyses [26–29]. Studies of physiological measurements of physical condition have typically been restricted to measures of urinary or faecal glucocorticoid metabolites [30,31], which indicate energy mobilization and allocation and allow for assessing periods of chronic, unhealthy stress, but are usually only very indirect measures of individual health and fitness itself. Non-invasive assessments of energetic condition in vertebrates have also included measurements of the urinary C-peptide of insulin [32,33], thyroid hormones [34,35] and isotopes [36,37]. Urinalysis dipstick tests to measure parameters such as the presence of leucoctyes, protein and blood in urine have also been used [38], although to our knowledge, without biological validation. Though useful in many aspects, these measures have usually been proposed for measuring other aspects of physical condition, such as energetic status, and either do not reflect the infectious status or degree of immune activation or inflammatory status of an individual at all, or do so only very crudely and/or indirectly.

More recently, in some studies of primates more direct data on the activation state of the immune system have been collected using the measurement of concentrations of faecal immunoglobulin A [39] as a measure of stress-related immune function and, in particular, urinary neopterin (uNEO, rhesus macaques [40,41]; Barbary macaques [14,42]; bonobos [43]; chimpanzees [44–46]), a by-product of macrophage activity and an early marker of the Th1 response of cell-mediated immunity [47]. One study has also evaluated the potential of faecal and urinary measurements of inflammatory acute phase proteins such as haptoglobin and C-reactive protein [40], but these analytes offer more limited promise. Given the usefulness of biomarkers that could be potentially measured non-invasively to reflect various aspects of health, and the dearth of such biomarkers that are available for use, new biomarkers of inflammation, infection and immune system activation that can be measured in the urine and/or faecal samples of large mammals are needed.

One possible candidate biomarker is the soluble form of the urokinase plasminogen activator receptor (suPAR), the binding site for the urokinase protein. The urokinase protein is an enzyme that is involved in the degradation of the extracellular matrix, making the uPA system central to processes such as fibrinoloysis, angiogenesis, cell proliferation and inflammation, and remodelling during tumours, metastasis and wound healing [48]. uPAR is predominantly expressed on immune cells (e.g. neutrophils, activated T-lymphocytes, macrophages), but may be cleaved from the cell surface, forming a soluble receptor, suPAR [49]. When inflammatory processes are activated by cytokines, uPAR expression is upregulated, increasing the levels of suPAR, too [50]. Usefully for animal field studies, in which recent food intake cannot be controlled, and in which samples must be collected opportunistically across the day, suPAR concentrations are unaffected by fasting state and diurnal variation [51,52]. In humans, concentrations of suPAR in blood are known to reflect both immune activation [53] and inflammation [54,55], and to correlate with elevated leucocyte counts and

concentrations of pro-inflammatory cytokines [52], as well as with inflammatory acute phase proteins such as C-reactive protein [54]. Plasma concentrations of suPAR are elevated in response to inflammatory conditions such as rheumatoid arthritis [54], and in response to different cancers [51,56,57] and immune disorders and infections [53,58]. High suPAR concentrations predict disease severity and mortality in patients with bacterial pneumonia [59], tuberculosis [60,61] and HIV [62]. Crucially, suPAR is measurable in urine, and in humans, concentrations in urine strongly correlate with those in blood [52,57,63]. However, to our knowledge, urinary suPAR has not been validated as a biomarker of immune activation and inflammation in species outside of humans. This is interesting given that genes orthologous to the human uPAR (PLAUR) gene have been found in several mammalian species (i.e. mouse, rat, dog, cow and primates), with many of the regulatory gene sites being evolutionarily conserved [64] suggesting similar biological functions across mammalian taxa.

In the present study, we assessed the validity of suPAR as a non-invasive urinary biomarker of inflammation and immune system activation processes in a primate model, the rhesus macaque (Study 1: biological and analytical validation). We took advantage of two infection experiments with the simian immunodeficiency virus (SIV) in combination with medical interventions (colon biopsy, bone marrow aspiration, surgery for lymph node removal) carried out as part of a separate study. Since elevated levels of suPAR reflect activation of immune and inflammatory systems (see above), including infection with the human immunodeficiency virus (HIV) [65], we predicted that SIV infection would result in an elevation of urinary suPAR concentrations. Furthermore, based on the finding that wounding and surgical trauma is associated with secretion of inflammation markers such as acute phase proteins [66,67], we also predicted that the medical interventions would activate suPAR secretion and that this would be reflected in elevated urinary suPAR concentrations.

We compare our results on urinary suPAR response to those of uNEO measured in the same samples to evaluate whether the two urinary biomarkers would respond similarly or differently to the immune and inflammation challenges. While we predicted that urinary suPAR levels would increase in response to both SIV infection and surgery-associated inflammation, we predicted that neopterin would not increase in response to inflammation caused by tissue trauma, since it is a more specific marker of cellular immune activation [47].

As the ultimate goal is to apply urinary suPAR measurements in studies of mammalian immune system activation and inflammation in the wild, we also evaluated the potential utility of urinary suPAR for use in field studies (Study 2: field validation). Specifically, we tested the effect of different conditions that are likely to be encountered by field researchers, especially those operating in remote and challenging natural environments [33,68] that are associated with sample collection, storage and processing (e.g. sample contamination, sample storage temperature, freeze–thaw cycles, etc. [26,43,69–74]). Since to the best of our knowledge, there have been no prior studies on the effect of the prescribed conditions on urinary suPAR concentrations, we generally made no specific predictions about the effects that such conditions might have. It has been reported though that suPAR concentrations in plasma are resistant to repeated freezing and thawing and that levels are stable for at least 72 h when stored at 4°C while storage at 20°C for 72 h resulted in increased suPAR concentrations [75]. Thus, we may expect urinary suPAR levels to be also more stable at a lower than ambient temperature and to resist repeated freezing and thawing.

# 2. Material and methods

## 2.1. Study 1: analytical and biological validation of suPAR measurements

### 2.1.1. Study animals and sample collection

For assessing the biological validity of urinary suPAR excretion in reflecting immune activation and inflammation in primates, we took advantage of two earlier collections of urine samples carried out between February and April 2014 and in October 2017 from adult rhesus macaques as part of two SIV infection experiments. The samples in 2014 were taken from six rhesus macaques (three males, three females; age: $4.6 \pm 0.2$ years) prior to and following an acute infection with SIV and were used previously for the validation of the measurement of uNEO [40]. Since all animals showed a strong neopterin response to the SIV infection, this situation provides also a useful test case for assessing the potential of urinary suPAR measurements in indicating immune status activation in response to an acute viral infection and to compare the response pattern to that of uNEO.

Samples collected in October 2017 came from seven male rhesus macaques (age: $3.7 \pm 1.3$ years) that had been infected with SIV 1 year prior to urine collection. At the time of sample collection, these animals underwent a combination of medical interventions, comprising colon biopsy, bone marrow aspiration and removal of peripheral lymph nodes. Since surgical trauma is known to induce inflammatory processes in the body (see Introduction), we took advantage of this situation for evaluating urinary suPAR's potential in reflecting the occurrence of inflammation in our study animals.

All study animals were housed individually in indoor cages at the German Primate Center under conditions described previously [40]. All urine samples were collected between 06.30 and 07.30 as reported earlier [40]. Only urine not contaminated with faeces was collected and samples were immediately protected from light. Samples were kept cold (4–7°C) upon collection and transferred to the endocrinology laboratory within 2 h of collection where they were vortexed, centrifuged and aliquoted before being stored frozen at −20°C until analysis.

From the SIV infection experiment in 2014 [40], a total of 100 urine samples were analysed. Specifically and similar to our previous neopterin measurements, we collected from each individual three samples taken prior to virus inoculation to establish pre-infection baseline suPAR levels. Following SIV infection, we collected 12–15 samples per individual (i.e. usually three samples per week) for 27–31 days to establish the suPAR response pattern to infection. From the animals in 2017, in total 67 urine samples were analysed. Here, we also analysed three urine samples (collected in the week preceding the medical interventions) to establish pre-surgery suPAR levels and daily samples (except for three animal days on which an animal did not urinate) for 7 days thereafter as inflammatory responses should be closely temporally linked to the interventions. Because the 2014 samples had been measured previously for neopterin (see above), these samples had been thawed and refrozen at least once prior to suPAR analysis. As a previous study indicated suPAR in blood samples to be resistant to repeated freeze–thaw cycles [75] and since our own experiments partly confirmed this result for urine samples as well (see below), we are confident that the thawing and refreezing process that the samples underwent did not affect suPAR's response pattern to infection markedly. Samples from 2017 were not used for any other analysis prior to the suPAR measurement, i.e. were kept frozen at −20°C all the time following their collection.

## 2.1.2. Analytical validation

In the absence of any published information on the measurement of urinary suPAR in macaques or any other non-human primate, we initially tested the general ability of two uPAR ELISA kits from RayBiotech (Norcross, USA) to detect suPAR in macaque urine. The two ELISAs were designed to measure uPAR in serum, plasma and cell culture supernatants of either rhesus macaques (Rhesus Macaque uPAR ELISA kit, Catalog #: ELK-uPAR) or humans (Human uPAR ELISA Kit, Catalog #: ELH-uPAR), but assays differed largely in their level of sensitivity (ELK-uPAR: 120 pg ml$^{-1}$; ELH-uPAR: 15 pg ml$^{-1}$). Both assays detected urinary suPAR in all test samples ($n = 6$; six animals). Concentrations measured by the two assays correlated strongly (Spearman rank correlation: $r = 1.0$, $p < 0.003$), and did not differ significantly (Wilcoxon signed-rank test: $T^{+} = 6$, $p = 0.4375$), indicating a similar level of cross-reactivity of the two suPAR antibodies with rhesus macaque urinary suPAR. Thus, both assays could principally be used to measure suPAR in urine of rhesus macaques. However, the eight times lower detection limit of the human uPAR ELISA allows detecting urinary suPAR at concentrations in the lower end range (i.e. between 15 and 120 pg ml$^{-1}$) where the macaque uPAR ELISA definitely fails. Also, due to the higher level of sensitivity, less volume of urine is needed for analysis using the human suPAR ELISA. This is particularly useful for field research on wild animals from which often only small amounts of urine are available [76,77], and in studies where the measurement of multiple analytes (e.g. steroid hormones, C-peptide, immune biomarkers) in the same sample is of interest [42]. We therefore routinely analysed all urine samples using the human uPAR ELISA, which is also more cost-effective due to its lower price.

To further examine the analytical validity of the human uPAR ELISA kit for the measurement of suPAR in macaque urine, we conducted: (a) a parallelism and (b) an accuracy test. For testing parallelism, we prepared a pooled sample which combined samples from males and females separately, and which contained high levels of suPAR. Samples were then serially diluted twofold from 1 : 5 to 1 : 160 in assay buffer and sample dilution curves and the assay standard curve were tested for differences between slopes [78]. Assay accuracy was assessed by spiking a urine pool sample diluted in buffer and containing low levels of endogenous suPAR, with known amounts of assay standard across the measurement range of the assay and determining the recovery of the added amounts.

### 2.1.3. Analysis of suPAR and neopterin in macaque urine

The urine samples collected for the biological validation (see above) were diluted 1 : 5–1 : 90 (depending on the suPAR concentration and amount of urine available) with assay buffer and 100 µl of the diluted urine was subsequently assayed following the manufacturer's instructions. All standards, samples and controls were measured in duplicate. From the 167 samples analysed, 17 (i.e. 10%) were below assay sensitivity. For the baseline samples of the two studies, 11.1% (2014) and 6.3% (2017) of the samples were below assay sensitivity with the 1 : 5 dilution used (i.e. around 90% of baseline samples were within the detection limit of the assay using a 1 : 5 dilution). For the samples below assay sensitivity, we assumed the minimum assay sensitivity value of 15 pg ml$^{-1}$ (cf. Higham *et al.* [33]). Since the actual values are known to be lower than this, this is a conservative approach that slightly reduces variation in the actual values in the data. We also confirmed that results were qualitatively the same if these values were not included. Inter-assay coefficients of variation (CV), calculated from the repeated measurement of a high- and low-value human urine sample run as quality controls in each assay, were 9.5% (high) and 12.0% (low), while mean intra-assay CV values were 1.2% (high) and 8.2% (low).

For comparison, we also measured uNEO. As mentioned above, samples collected for the SIV infection experiment in 2014 had been analysed earlier [40] and data were taken from these previous measurements. In this study, therefore, only samples collected from animals that underwent the medical interventions in 2017 were analysed for uNEO. The uNEO measurement was carried out in five of the seven study subjects using 1 : 25 diluted urine samples assayed in the IBL Neopterin ELISA (Art. No. RE59321; IBL International GmbH, Hamburg, Germany) as described previously [40]. Inter-assay CV values for high- and low-value quality controls run in each assay were 10.1% (high) and 11.6% (low), while mean intra-assay CV values were 8.8% (high) and 7.1% (low). To adjust for differences in the volume and concentration of the collected urine, suPAR and uNEO values were indexed to the level of urinary creatinine (Cr) measured by the method described by Bahr *et al.* [79] and suPAR and uNEO concentrations are presented as ng/mg Cr.

Although we did not also measure specific gravity (SG) for all biological validation samples, we measured 28 of the samples for SG, and found that the correlation between Cr values and SG values was 0.97, suggesting that indexing by SG would have produced almost identical results in this case.

### 2.1.4. Data analysis

Data are available on datadryad [80].

Differences between slopes of sample dilution curves with the respective suPAR standard curve were tested using a Tukey test. Comparison between added and recovered amounts of suPAR in the accuracy experiment was carried out using the Spearman rank correlation test.

We generated composite (average) profiles of urinary suPAR and uNEO response patterns to SIV infection and surgery-associated inflammation, respectively. For this, we calculated the percentage change in suPAR and uNEO levels (relative to the mean pre-infection/pre-surgery baseline levels; set as 100%) for each sampling day following the respective treatment for each individual and averaged values across all individuals. We assessed the temporal course of suPAR and uNEO excretion by determining the lag time in days from the day of administration of SIV or the day of the medical interventions, respectively, to (i) the day of first increase in suPAR and uNEO levels above a threshold value determined as the mean + 2 s.d. of the pre-infection/pre-surgery baseline levels indicative of a significant rise [81] and (ii) the day of peak response, separately for each individual. To assess the magnitude of the two immune marker responses to the immune challenges, we also determined for each study animal the suPAR and uNEO peak-to-baseline (P/B) ratio.

## 2.2. Study 2: field validation

### 2.2.1. Study animals and sample collection

This study was conducted in January/February 2019 using urine samples collected from eight individually housed healthy adult non-infected male rhesus macaques of the German Primate Center's rhesus macaque colony not allocated to any experimentation. Housing conditions of the animals were as described for animals used in Study 1. From each study animal, a 4–5 ml urine sample was collected between 06.00 and 07.00 as described for Study 1. Samples were brought to the

endocrinology laboratory within 2 h of collection where they were processed for the different experimental treatments as outlined below.

### 2.2.2. Urinary suPAR, creatinine and SG analyses

Urinary suPAR concentrations were measured using the human uPAR ELISA kit validated in Study 1. Prior to assay, all urine samples were diluted 1:5 with assay buffer to bring the samples into the working range of the assay. Inter-assay CV, determined as described for Study 1, were 8.2% and 11.1% for high- and low-value quality controls, respectively.

To adjust suPAR levels for differences in urine volume and concentration, we examined the performance of two methods of adjustment: indexing to urinary creatinine (Cr), the most common approach to normalize analyte concentrations in urine [82], and correcting by urinary SG, a more recent used alternative to creatinine adjustments [83–85]. Indexing urinary analyte concentrations with SG rather than Cr may be particularly advantageous for indexing urine samples from wild animals [68,74]. Because Cr is a by-product of muscle activity, it can be influenced by a multitude of factors, such as sex and age [86,87], physical activity [88], muscle mass [86,89] and diet [85,90] and some of these factors, in particular physical activity and diet, can vary substantially across wild-living individuals. As a chemical compound, Cr is also potentially sensitive to degradation under unfavourable environmental conditions of sample processing, storage and shipment [33,91] that are often encountered by field researchers when collecting and processing urine samples in the field [68,74]. By contrast, SG, which represents the density of urine relative to water, appears to be less affected by the factors that potentially influence Cr [85,87] and thus SG has been proposed to be a more reliable and tolerant urine concentration–dilution indicator [83–85,92]. Both Cr and SG were initially measured in fresh samples and afterwards in all treatment samples (including controls) to assess the effect of treatment on Cr and SG values.

As researchers using field samples would measure Cr or SG along with urinary suPAR concentrations from the same sample, we took the Cr and SG value from the same sample as the urinary suPAR concentration in each experimental treatment, rather than from controls. As such, our results show how indexed urinary suPAR concentrations (those that would be of interest to field researchers) respond to the different treatments. We measured Cr as described by Bahr *et al.* [79] and determined SG using a digital hand-held refractometer (PAL-10S; Atago Inc., Bellevue, USA). For Cr, intra- and inter-assay CV calculated from low- and high-value quality controls run in each assay were less than 5%, while CV values for SG measurements were both less than 0.5%. Urinary suPAR concentrations were calculated as urinary suPAR (ng mg$^{-1}$ Cr) or urinary suPAR (ng ml$^{-1}$) corrected for SG (corr. SG). The correction with SG was carried out using the formula described by Miller *et al.* [83], with the SG population average for the macaque urine test samples being 1.0196.

As Cr and SG are also widely used when indexing other urinary analytes such as steroid hormones, C-peptides or neopterin [33,82,84], we also assessed the effects of the different treatments on Cr concentrations (as mg ml$^{-1}$ urine) as well as on SG separately.

### 2.2.3. Contamination with soil and faeces

Since urine samples collected from wild animals may potentially be contaminated with soil or faecal matter [69], we tested for a potential effect of this contamination on indexed urinary suPAR concentrations. For this, 250 µl urine of each sample was pipetted into polypropylene tubes which contained 55 ± 5 mg of soil or freshly collected rhesus macaque faeces, representing a proportionally substantial contamination. Each contaminated sample was briefly vortexed and incubated for 5 min at room temperature before samples were centrifuged (1 min at 3000 r.p.m.) and the supernatant was transferred into a clean tube. Tubes containing 150 µl of urine from the same samples were prepared as matched controls. All samples were stored in the refrigerator (4–6°C) for 5 h thereafter (to mimic storage in a cooler in the field) and subsequently at −20°C until analysis for suPAR, Cr and SG.

### 2.2.4. Exposure to sunlight

Since collection of urine samples from wild animals is often carried out in tropical areas where exposure of samples to direct sunlight either at times of collection or processing is likely and potentially unavoidable, and since neopterin, another biomarker of immune activation, is known to degrade quickly when exposed to sunlight [43,93], we evaluated the effect of exposing rhesus urine to direct sunlight on suPAR, Cr and SG

measures. For this, aliquots of 150 µl of urine from each test sample were placed for 3 h in direct sunlight (samples were placed on an icepack to exclude potential temperature effects [43]). As matched controls, sets of tubes with 150 µl of urine were placed for 3 h in the shade at room temperature. All samples were frozen at −20°C thereafter until analysis for suPAR, Cr and SG.

### 2.2.5. Sample storage at room and cool temperatures

Since urinary compounds can be subject to degradation when stored at above 0°C temperatures for prolonged periods [69,94], and to mimic a situation where urine samples can only be kept frozen after a certain amount of time upon their collection, we tested the vulnerability of urinary suPAR, Cr and SG to storage of rhesus urine at (i) ambient, i.e. room temperature (21–22°C) and (ii) at 5–7°C (refrigerator) for several days before freezing them. Specifically, for each treatment urine samples were portioned into four aliquots of 200 µl each. While one set of samples was immediately stored frozen (matched control samples), all other samples were placed in the dark for 2, 5 and 9 days at the temperatures mentioned above. After the respective storage times, samples were also placed at −20°C until analysis for suPAR, Cr and SG together with the matched controls.

### 2.2.6. Freezing and 24-h freeze–thaw cycles

To evaluate the effect of repeated freeze–thaw cycles and storing samples in-between at ambient temperature for 1 day (as to simulate power loss or inconstant electricity supply of a freezer, or thawing during transport for analyses) on urinary suPAR, Cr and SG measures, we pipetted 500 µl from each urine sample into polypropylene cups and stored these frozen at −20°C. Additional tubes containing 150 µl of urine from the same samples were also stored frozen as matched controls. After 5 days stored frozen, test samples were thawed and stored in the dark at room temperature (RT) for 24 h. From each sample 150 µl were stored at −20°C together with the remaining urine (first freeze–thaw cycle). The procedure was repeated twice to provide samples that had been thawed and refrozen two or three times, respectively (second and third freeze–thaw cycle).

In addition, we assessed separately whether one freezing–thawing–refreezing situation with a short 4 h post-thaw storage at RT, a situation often encountered in the laboratory when, for example, samples have to be re-measured or are re-used for analysis of additional compounds, would affect suPAR, Cr and SG measures. All test samples were finally stored frozen at −20°C until analysis for suPAR, Cr and SG together with the matched controls.

### 2.2.7. Lyophilization

Lyophilization (freeze-drying) is known to stabilize sensitive compounds, such as hormones and immune markers in biological samples for prolonged periods of time [33,72,74]. We tested the recovery of urinary suPAR, Cr and SG after lyophilization of urine, preparing aliquots of 200 µl of each test sample in polypropylene cups. Tubes containing 150 µl of urine from the same samples were prepared as matched controls. Test samples and controls were then stored frozen at −20°C for two weeks. All test samples were then freeze-dried for 6 h at −20°C and stored frozen again thereafter. Prior to analysis, lyophilized samples were reconstituted in 200 µl Millipore purified water. Samples were vortexed twice for 15 s each and kept at RT in the dark for 30 min. Samples were then analysed for urinary suPAR, Cr and SG together with the matched controls.

### 2.2.8. Data analysis

As sample sizes were small, we used non-parametric statistics. Statistical tests regarding Cr and SG measurements were performed on raw values (mass per volume, e.g. mg ml$^{-1}$), whereas statistical analyses of suPAR results were carried out on concentrations corrected for SG (see Results). We used the Wilcoxon signed-rank test to assess the effects of a single treatment, and the Friedman test for experiments containing repeated measures over time. In these latter cases, we undertook post hoc paired exact Wilcoxon signed-rank tests to determine the period after which effects on values over time first became significant. For all treatments, we undertook Spearman rank correlation tests to investigate whether treated and control values were correlated irrespective of whether the treatment resulted in a significant change in urinary Cr, SG and suPAR concentrations or not. All statistics were undertaken in Sigma Stat 13.0 (Systat Software GmbH, Erkrath, Germany). Two-tailed tests using exact probabilities were performed, with analyses considered significant when $p < 0.05$.

# 3. Results

## 3.1. Study 1: analytical and biological validation of suPAR measurements

### 3.1.1. Analytical validation

Displacement curves of serially diluted pooled urine samples of both male and females run parallel to the suPAR standard curve, with no differences in slopes between displacement curves and standard curve (Tukey tests, male urine pool: $t = 1.3715$; d.f. = 9; $p = 0.2034$; female urine pool: $t = 1.4423$; d.f. = 9; $p = 0.1831$). Mean recovery of known suPAR concentrations added to urine was $45.5 \pm 4.7\%$ (mean ± s.d.; range 39.9–50.0%). Because the recovery was unexpectedly low, the accuracy experiment was repeated, and a control experiment in which buffer samples rather than urine samples were spiked with suPAR standard was carried out alongside. The recovery from the spiked urine samples was again low ($42.0 \pm 3.2\%$), while recovery of the spiked buffer samples was high ($90.3 \pm 2.7\%$) and in the expected range. Although overall recovery of suPAR in urine was thus lower than expected, crucially recovery values were consistent across the five doses tested, with a correlation coefficient between added and recovered amounts of $r_s = 1.0$.

### 3.1.2. Biological validation: suPAR response to SIV infection

The variation in individual mean pre-treatment baseline values ranged between 0.33 and 0.94 ng mg$^{-1}$ Cr, and sex-specific averages were: males, $0.57 \pm 0.32$ ng mg$^{-1}$ Cr; females, $0.42 \pm 0.14$ ng mg$^{-1}$ Cr. All six study subjects showed a strong increase (4.5- to 31.0-fold; median: 8.3-fold) in urinary suPAR concentrations following SIV virus infection. Median responses by sex were: males, 10.7-fold increase; females, 7.9-fold increase. The first significant elevation in suPAR concentrations was usually detected at day 13 post-virus inoculation and peak concentrations were recorded on average 2 days later (table 1 and figure 1). All study animals underwent medical interventions (colon biopsy, bone marrow aspiration and peripheral lymph node removal) on either day 14 or day 15 of the experiment. In all animals, the suPAR rise occurred in urine samples collected prior to the interventions, and in 4/6 animals even peaked before these interventions, indicating that suPAR excretion was induced independent of the latter. Urinary suPAR concentrations returned to baseline concentrations within 5–9 days following peak concentrations (figure 1). In comparison, uNEO responded faster and more gradually, but also more markedly (10.5- to 29.7-fold, median: 16.8-fold) to the SIV administration in each individual (table 1 and figure 1). Similarly to suPAR concentrations, uNEO concentrations declined after day 14/15 post-infection, but contrasting to suPAR concentrations, uNEO concentrations stayed elevated above pre-treatment levels until the end of the sampling period four weeks after treatment (figure 1).

### 3.1.3. Biological validation: suPAR response to medical interventions

The variation in individual mean pre-treatment baseline values ranged between 0.27 and 1.19 ng mg$^{-1}$ Cr. All seven study males showed a strong response in urinary suPAR levels to the medical interventions undertaken, with suPAR concentrations increasing 6.0- to 18.7-fold (median: 6.3-fold) compared to pre-treatment baseline values (table 1 and figure 2). In all animals, the rise in suPAR concentrations occurred immediately, i.e. highest levels of suPAR were found in the first urine sample collected on day 1 post-intervention. suPAR levels gradually declined thereafter with concentrations being close to pre-intervention baseline levels by day 7 (figure 2). By contrast, uNEO responded only weakly to the interventions conducted. Specifically, uNEO concentrations increased, on average, only 1.5-fold (range: 1.3- to 3.6-fold) over pre-treatment baseline values (table 1 and figure 2), with these slightly elevated levels being maintained for the entire 7 days on which samples were collected post-intervention.

## 3.2. Study 2: field validation

### 3.2.1. Creatinine and SG measurements

In general, Cr concentrations were only weakly affected by the different experimental treatments, with changes in Cr concentrations in response to any treatment usually not exceeding 5% of the respective control values (data not shown). Such a small change is well within typical measurement variation and error (i.e. within assay variability) and thus is unlikely to have major effects on Cr-indexed

**Table 1.** suPAR and neopterin (uNEO) concentrations in urine in response to SIV infection and medical interventions in individual rhesus macaques.

| animal ID | suPAR | | | | | uNEO | | | | |
|---|---|---|---|---|---|---|---|---|---|---|
| | Pre[a] | peak[b] | P/B ratio[c] | first rise[d] | peak[e] | Pre[a] | peak[b] | P/B ratio[c] | first rise[d] | peak[e] |
| **SIV infection** | | | | | | | | | | |
| Male 2583 | 0.46 | 4.92 | 10.7 | +14 | +15 | 0.11 | 2.50 | 22.7 | +9 | +15 |
| Male 14893 | 0.94 | 6.14 | 6.5 | +13 | +15 | 0.21 | 3.45 | 16.4 | +9 | +15 |
| Male 15226 | 0.33 | 10.22 | 31.0 | + 9 | +15 | 0.18 | 3.08 | 17.1 | +9 | +14 |
| Female 14875 | 0.34 | 2.69 | 7.9 | +13 | +14 | 0.19 | 1.99 | 10.5 | +11 | +14 |
| Female 14892 | 0.58 | 5.06 | 8.7 | +13 | +15 | 0.13 | 3.86 | 29.7 | +3 | +15 |
| Female 2503 | 0.35 | 1.58 | 4.5 | +13 | +15 | 0.21 | 2.72 | 13.0 | +11 | +14 |
| median | 0.41 | 4.99 | 8.3 | +13 | +15 | 0.19 | 2.90 | 16.8 | +9 | +15 |
| **Interventions** | | | | | | | | | | |
| Male 2729 | 1.19 | 7.03 | 5.9 | +1 | +1 | 0.22 | 0.33 | 1.5 | n.d. | n.d. |
| Male 2772 | 1.09 | 6.92 | 6.3 | +1 | +1 | 0.21 | 0.40 | 1.9 | n.d. | n.d. |
| Male 14954 | 0.27 | 2.88 | 10.7 | +1 | +1 | —[f] | — | — | — | — |
| Male 15899 | 1.13 | 7.08 | 6.3 | +1 | +1 | — | — | — | — | — |
| Male 15925 | 1.07 | 20.06 | 18.7 | +1 | +1 | 0.50 | 0.68 | 1.4 | n.d. | n.d. |
| Male 15926 | 0.55 | 3.28 | 6.0 | +1 | +1 | 0.20 | 0.25 | 1.3 | n.d. | n.d. |
| Male 15933 | 0.40 | 3.84 | 9.6 | +1 | +1 | 0.10 | 0.36 | 3.6 | n.d. | n.d. |
| median | 1.07 | 6.92 | 6.3 | +1 | +1 | 0.21 | 0.36 | 1.5 | n.d. | n.d. |

[a]Pre-treatment concentrations (mean) in ng mg$^{-1}$ Cr (suPAR) or µg mg$^{-1}$ Cr (uNEO).

[b]Peak concentrations in response to SIV infection or surgery in ng mg$^{-1}$ Cr (suPAR) or µg mg$^{-1}$ Cr (uNEO).

[c]Peak-to-baseline ratio, i.e. x-fold increase of peak concentrations above mean pre-treatment concentrations.

[d]Lag time in days between SIV administration or surgery and first significant rise in suPAR concentrations.

[e]Lag time in days between SIV administration or surgery and peak suPAR and uNEO response.

[f]— = no data

[g]Not determined because uNEO responded only weakly (see text and figure. 2).

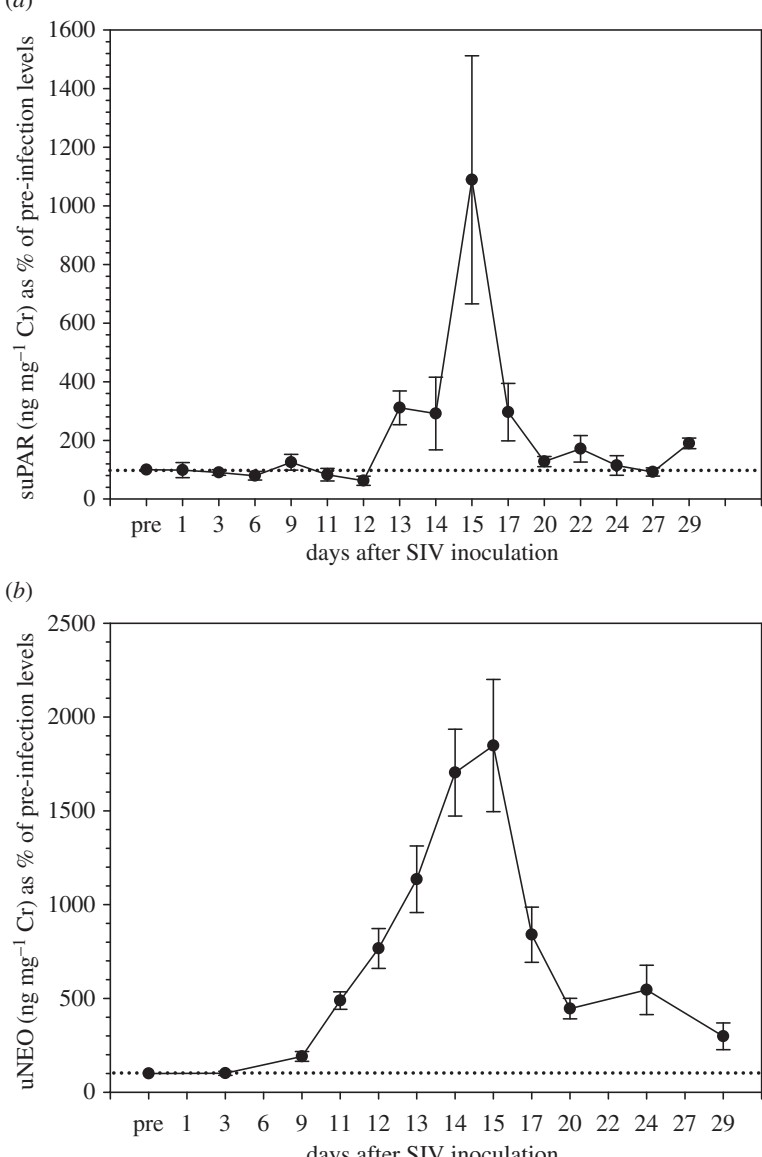

**Figure 1.** Percentage of response in (*a*) urinary suPAR and (*b*) uNEO concentrations to SIV infection in rhesus macaques. Data points represent mean ± s.e.m. values calculated across the six individuals studied. Percentages were calculated in relation to pre-SIV inoculation baseline values (pre = 100%; dotted line).

results [74]. The only treatments that caused changes in Cr concentrations of greater magnitude were contamination of the urine with faeces and storage of the samples at room temperature for several days. Contamination with faeces resulted in a decline in Cr levels to $90.6 \pm 0.9\%$ of control values, which represents a statistically significant decline ($T^+ = 0$, $p = 0.0078$). Similarly, storage of urine at ambient temperature for 9 days caused a decline in Cr concentrations to $85.2 \pm 3.0\%$, again a statistically significant change ($T^+ = 1$, $p = 0.0156$). Nevertheless, Cr values following these two experimental treatments (as well as Cr values of all other treatments) remained strongly and significantly correlated with control values (all $r_s > 0.90$; all $p < 0.0001$). In contrast to Cr, SG values were extremely stable across all treatments, with individual SG values in 94.2% of the samples differing no more than by ±0.001 relative to controls, a difference which reflects the measurement inaccuracy of the refractometer. Because SG was thus much less affected by our experimental treatments than Cr, we report all results on urinary suPAR in response to the different treatments as suPAR concentrations (ng ml$^{-1}$) corrected for SG. Given the extremely high stability of SG across treatments, any significant change in suPAR levels found will thus directly mirror the chemical or structural instability of suPAR under the conditions tested.

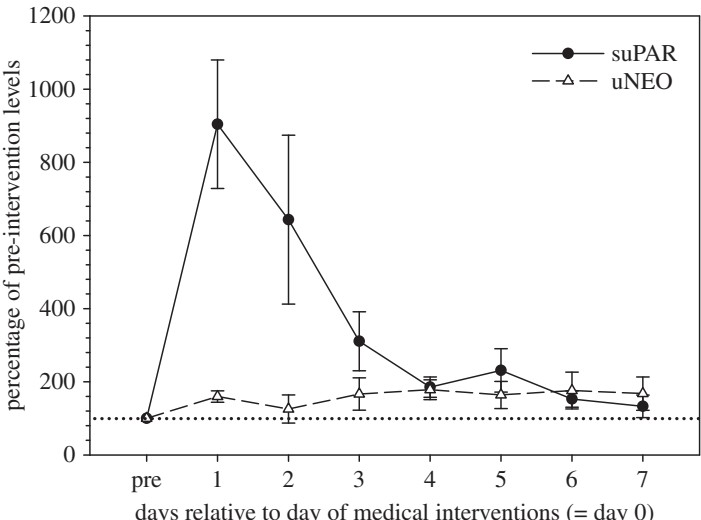

**Figure 2.** Percentage of response in urinary suPAR and uNEO concentrations to medical interventions (see text for details) in rhesus macaques. Data points represent mean ± s.e.m. values calculated across the individuals studied ($N = 7$ for suPAR, $N = 5$ for uNEO). Percentages were calculated in relation to pre-intervention baseline values (pre = 100%; dotted line).

Table 2 presents a summary of the urinary SG-corrected suPAR results which, in the following, are considered in more detail.

### 3.2.2. Contamination with soil and faeces

Both contamination of urine with soil and faeces affected urinary suPAR concentrations substantially. suPAR levels declined to values of $44.4 ± 3.6\%$ (soil) and $73.5 ± 4.4\%$ (faeces) of controls (figure 3a), which in both cases represents a statistically significant reduction (table 2). Despite these marked contamination-induced changes, for both cases ranks of suPAR concentrations nevertheless correlated strongly and significantly to those of controls (table 2), reflecting that the rank order of the different individual samples (from high to low) remained largely unchanged (figure 3b,c).

### 3.2.3. Exposure to sunlight

Placing samples for three hours into direct sunlight resulted in a decline of urinary suPAR concentrations to values of $75.4 ± 3.1\%$ of controls, representing a statistically significant reduction (table 2). Nevertheless, rank suPAR concentrations correlated very strongly to those of control values (table 2), reflecting that the rank order of the individual samples remained largely unchanged.

### 3.2.4. Sample storage at room and cool temperatures

Storing urine samples at either room temperature (RT, 21–22°C) or in the refrigerator (5–7°C, fridge) for up to 9 days both resulted in a gradual decrease in urinary suPAR concentrations over time (figure 4a,c). The magnitude of this temporal decline in suPAR concentrations was clearly temperature related, with stronger effects recorded for samples stored at the higher temperature (figure 4a,c). For both storage conditions, the decreases in suPAR concentrations were highly significant over the whole experimental period (table 2). Post hoc Wilcoxon signed-rank tests showed that for both conditions the decline in suPAR levels was already significant at day 2 of storage (RT: $T^+ = 0$, $p = 0.0078$; fridge: $T^+ = 2$, $p = 0.0234$). However, while the suPAR decline in samples stored for two days at RT was already marked, resulting in values of only $70.1 ± 10.1\%$ of controls, the decline in samples stored at 5–7°C was relatively small, with values representing $89.9 ± 3.1\%$ of controls (figure 4a,c). Accordingly, rank suPAR concentrations in samples stored in the fridge for 2 (and also 5) days were strongly correlated to those of controls (both $r_s = 0.98$, $p < 0.0001$), reflecting that the rank order of the individual samples remained largely unchanged at both 2 and 5 days (but not at 9 days) of storage (figure 4b). By contrast, suPAR levels of RT-stored samples did not correlate significantly with those of controls at either time point ($r_s = 0.04$–$0.67$, $p = 0.06$–$0.88$), reflecting the fact that declines were sufficiently variable to change the rank order of the samples substantially (figure 4d).

**Table 2.** Summary of urinary suPAR (ng ml$^{-1}$) corr. SG results. $N = 8$ for all treatments. Values in italics indicate a statistically significant difference.

| treatment group | treatment | values as % of control (mean ± s.e.m.) | Friedman test | | Wilcoxon test | | Spearman correlation against control | |
|---|---|---|---|---|---|---|---|---|
| | | | $\chi^2$ | $p$ | $T^+$ | $p$ | $r_s$ | $p$ |
| collection and storage issues | contamination with soil | 44.4 ± 3.6 | | | 0 | *0.0078* | 0.91 | *<0.0001* |
| | contamination with faeces | 73.5 ± 4.4 | | | 0 | *0.0078* | 0.86 | *<0.002* |
| | exposure to sunlight | 75.4 ± 3.1 | | | 0 | *0.0078* | 0.98 | *<0.0001* |
| | 9 days storage in fridge | 73.6 ± 5.8 | 14.8500 | *0.0019* | | | 0.86 | *<0.002* |
| | 9 days storage at room temperature | 15.3 ± 6.9 | 22.3077 | *< 0.0001* | | | 0.04 | 0.8849 |
| processing issues | three freeze/thaw cycles with 24 h post-thaw | 65.9 ± 9.2 | 16.6500 | *0.0008* | | | 0.26 | 0.4979 |
| | one freeze/thaw cycle with 4 h post-thaw | 96.5 ± 3.3 | | | 14 | 0.6406 | 0.95 | *<0.0001* |
| | lyophilization | 110.3 ± 4.1 | | | 30 | 0.1094 | 0.98 | *<0.0001* |

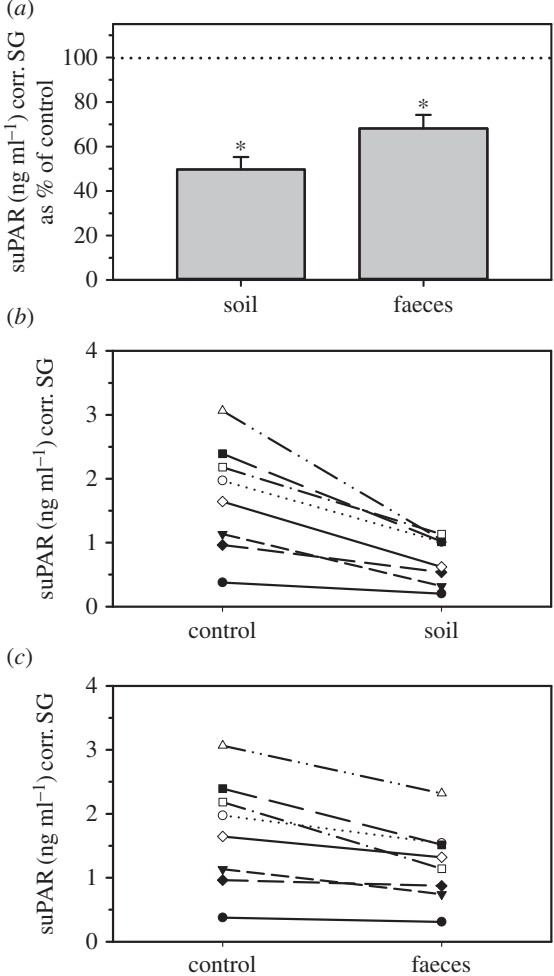

**Figure 3.** Effect of soil and faecal contamination of urine samples on urinary suPAR. Urinary suPAR concentrations are shown as (*a*) percentage (mean ± s.e.m.) of control values (100%, dotted line), and absolute concentrations after contamination with (*b*) soil and (*c*) faeces. For each treatment, asterisks indicate concentrations that differ significantly from controls.

### 3.2.5. Freezing and 24-h freeze–thaw cycles

Urinary suPAR concentrations were significantly affected by freeze–thaw situations. Samples exposed to several 24-h freeze–thaw cycles declined gradually in suPAR values, with declines getting more pronounced with increasing numbers of freeze–thaws (figure 5*a*). These declines were highly significant when all three freeze–thaw cycles were considered (table 2). Post-hoc Wilcoxon signed-rank tests showed that the decline in suPAR concentrations was already statistically significant after just one freeze–thaw situation resulting in values of $84.5 \pm 5.1\%$ of controls ($T^{+} = 1$, $p = 0.0156$; figure 5*a*). However, rank concentrations after just one freeze–thaw cycle were strongly correlated to those of controls ($r_s = 0.98$; $p = 0.0001$), reflecting the fact that the relative rank order of the samples remained largely unchanged (figure 5*b*). Urinary suPAR concentrations in samples subjected to two freeze–thaw cycles declined to $72.5 \pm 7.1\%$ of controls ($T^{+} = 1$, $p = 0.0156$; figure 5*a*), but were still significantly correlated to those of controls ($r_s = 0.69$; $p = 0.0474$). These declines were sufficiently variable, however, to change the rank order of samples markedly (figure 5*b*). Exposing samples to three freeze–thaw situations resulted in the strongest reduction in suPAR levels ($65.9 \pm 9.2\%$ of controls; $T^{+} = 0$, $p = 0.0078$; figure 5*a*) and rank sample concentrations were no longer significantly correlated with those of controls (table 2). By contrast, in samples subjected to one freezing–thawing–refreezing situation with a short post-thaw period following the initial thaw, suPAR concentrations remained stable ($96.5 \pm 3.3\%$ of controls; figure 5*c*) and did not differ significantly from control values (table 2). Also, rank suPAR concentrations following this condition were strongly correlated to those of controls (table 2), reflecting the fact that the relative rank order of the samples remained largely unchanged (figure 5*c*).

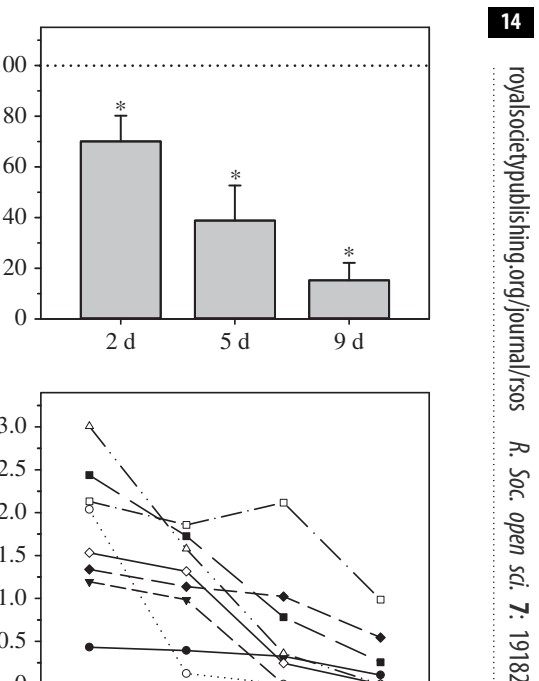

**Figure 4.** Effect of fridge- (left graphs; *a,b*) or room temperature storage (right graphs; *c,d*) on urinary suPAR concentrations. Urinary suPAR concentrations are shown as percentage (mean ± s.e.m.) of control values (100%, dotted line; *a,c*), and absolute concentrations (*b,d*). For each treatment, an asterisk indicates concentrations that differ significantly from controls. d = days of storage.

### 3.2.6. Lyophilization

Subjecting urine samples to lyophilization did not change urinary suPAR concentrations significantly (110.3 ± 4.1% of controls; table 2), and due to low variation in changes as a result of this treatment, rank suPAR concentrations were very strongly correlated with those of controls, reflecting the fact that the relative rank order of the samples remained largely unchanged (table 2).

## 4. Discussion

Our study examined whether suPAR measured from urine is an appropriate biomarker for assessing infections, inflammation and tissue damage in a non-human primate species. Using rhesus macaques as our model, we found that a commercial ELISA designed for the measurement of human uPAR is analytically suitable for the analysis of suPAR in the urine of macaques. As predicted, our biological validation shows that suPAR measurements in the rhesus macaque, like in humans, do reliably reflect both infectious status, rising in response to SIV infection, and inflammation, rising in response to surgical tissue trauma, allowing the study of immune system activation and inflammation in species other than humans. Finally, our field validation demonstrates that urinary suPAR is quite robust to a number of the issues associated with field collection, processing, and storage, and as long as access to a freezer is possible, is a suitable health marker for field studies. Given that uNEO concentrations showed a stronger and broader response to infection than suPAR, but that suPAR concentrations showed a clear response to tissue inflammation whereas uNEO did not, it seems that NEO is likely to be a better marker for measuring infection (particularly those induced by intracellular pathogens), and suPAR for tissue damage and inflammation (which may include inflammatory responses to infection).

### 4.1. Analytical validation

We validated a uPAR assay originally developed to quantify uPAR in human serum and plasma for the measurement of suPAR in macaque urine. Our analytical validation revealed that the human uPAR assay clearly recognizes urinary macaque suPAR and that suPAR can be measured reliably from macaque urine samples over a large dilution range. However, the lower than expected recovery of suPAR

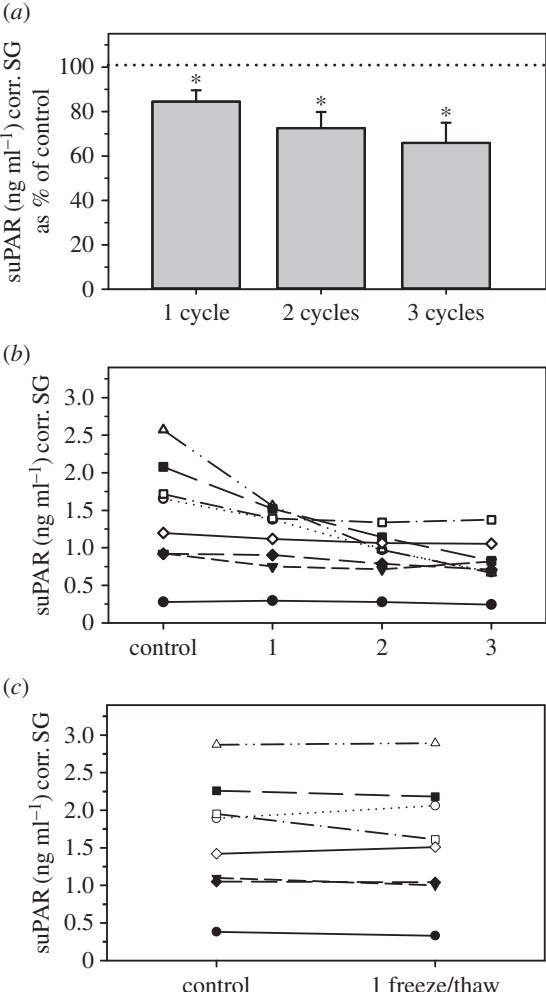

**Figure 5.** Effect of freeze/thaw cycles on urinary suPAR concentrations. Samples were either subjected to successive freezing–thawing–refreezing situations, each including a 24 h post-thaw storage at RT (*a*, *b*), or were subjected to one freezing–thawing–refreezing situation with a 4 h post-thaw storage at RT (*c*). Urinary suPAR concentrations are shown as (*a*) percentage (mean ± s.e.m.) in comparison to controls (=100%) and (*b*,*c*) absolute concentrations. Asterisks indicate concentrations that differ significantly from controls.

added to urine suggests that the urinary matrix may interfere with antibody binding as has been reported for the cytokine macrophage migration inhibitory factor measured from human plasma [95]. The results of our parallelism test indicate that the amount/proportion of urine in the sample (and thus the amount of the substance that is interfering in the assay matrix and reducing the recovery rate) does not affect the strength of the interference, and samples diluted perfectly parallel to the standard curve. Thus, the relative differences in suPAR levels between samples remain largely unaffected by the lower recovery, thereby providing robust data for intra- and inter-animal comparisons, irrespective of absolute values measured. The general robustness of the suPAR measurements is also confirmed by the results from our biological validation experiments (see below).

## 4.2. Biological validation of urinary suPAR measurements

By using both infections and medical interventions that resulted in tissue damage and inflammation, and by testing both urinary suPAR and uNEO concentrations in parallel, we were able to assess the specific aspects of condition about which suPAR might be most informative. As predicted, both uNEO and suPAR rose in response to SIV infection indicating that, similar to humans [55], suPAR has potential as a biomarker of infection also in the rhesus macaque. Nonetheless, the less marked and less-sustained suPAR response to SIV infection when compared with uNEO, suggests that the latter might be a better choice for infection studies, at least for those studying infections inducing a cell-mediated

(Th1-type) immune response to intracellular acting pathogens, like SIV [47,96]. Neopterin is not responsive to infections induced by extracellular pathogens (e.g. multicellular parasites, extracellular acting bacteria) though which stimulate humoral immunity via an antibody-mediated and anti-inflammatory Th2-type response [97]. In such cases, monitoring urinary suPAR may have advantages over the measurement of uNEO given that increased systemic levels of suPAR are found in human individuals suffering from viral, bacterial and parasitical infections [55]. Urinary suPAR thus may be a less specific and more general marker of an animal's infectious state compared to uNEO, but future studies need to verify this.

In contrast and as predicted, uNEO concentrations showed little response to invasive surgery and associated tissue damage, whereas urinary suPAR concentrations increased very strongly. Since other inflammation markers such as the acute phase proteins haptoglobin and C-reactive protein have only very little value when measured from urine or faeces [40], suPAR may be a more sensitive and more reliable alternative for diagnosing and monitoring inflammatory processes non-invasively in primates and potentially other mammals as well. In addition, since urinary suPAR showed responsiveness to infection, injuries and inflammation, it might be taken as a general measure of certain aspects of animal health, since researchers will rarely know what particular infection or issue might be impacting an animal's health status, making a marker that captures both infections and types of inflammation especially useful. We also envisage that the assay described here has the potential to detect small-scale differences in baseline immunity in non-infected animals given the low sensitivity of the assay and our finding that pre-infection/inflammation baseline values show a three- to fivefold variation between different individuals. In this respect, monitoring suPAR may also aid in studies investigating the links between sociality, health and fitness in wild-living animals, a research topic of growing interest [13,14,98]. Although we did not set out to test the relationship between uNEO and suPAR concentrations, and do not have sufficient sample sizes to investigate this, a Pearson correlation between the two value sets produced $r = 0.64$ for the samples collected in 2014 during the SIV infection study, and $r = 0.51$ for the samples collected in 2017 for the surgery study—both values are highly significant. Nonetheless, this analysis is pseudo-replicated because samples come from the same individuals, and so should be treated with caution. A more complete analysis of the relationship between the two variables should be conducted.

## 4.3. Field validation

The potential application of urinary suPAR measurements in studies of free-ranging animals depends largely on the stability of the compound under ambient, often unfavourable, field conditions. It is therefore essential to explore the effect of sample handling, processing and storage issues given that urinary analytes are often vulnerable to degradation [33,70]. Similarly important is to assess the reliability of creatinine (Cr) and/or SG measures required to correct urinary analyte concentrations for urine dilution [68,83,92]. Here, we could show that SG is not affected at all by the conditions tested whereas Cr, although being largely robust as well (see also [69,74]), was impacted by faecal contamination and storage of samples at elevated temperatures for several days. Thus, SG has clear advantages over Cr for normalizing urinary analyte concentrations, at least under suboptimal conditions of sample collection and handling, confirming results from other studies in which both measures have been compared [84,85,92].

Our field experiments (Study 2) also show that suPAR is at least partially robust to issues associated with fieldwork. That said, there are a number of issues that researchers must bear in mind. Firstly, soil and faecal contamination both reduce urinary suPAR concentrations substantially, and should be avoided wherever possible. Unlike for some other urinary compounds such as the C-peptide of insulin [69], it is soil, rather than faecal contamination, which has the bigger impact. This creates much bigger issues for some taxa than others. For example, urine from terrestrial species, such as ungulates, is likely to need to be collected from the ground, unlike urine from arboreal primates such as orangutans or chimpanzees, which can be collected directly mid-stream from trees or from vegetation [37,99]. In studies that have collected urine from the ground, it is possible to use squeezy 1 ml disposable micropipettes and transfers to other microcentrifuge collection tubes, to clean urine of soil which precipitates within a few minutes. We recommend such cleaning here if soil contamination is unavoidable. Urine can be also collected from the ground using Salivette swabs which helps to provide a cleaner sample since any dirt will adhere to the outer part of the swab [76]. Nonetheless, our results show that the rank order of samples from high to low is largely unaffected by such contamination, allowing relative between-sample differences in concentrations of urinary suPAR be

assessed reliably even when contamination cannot be completely avoided. Similarly, long-term exposure to direct sunlight also causes a substantive reduction to urinary suPAR concentrations, probably a result of UV radiation acting on the suPAR molecule as has been reported for uNEO [93]. Extended sunlight exposure during both sample collection and analysis should therefore also be minimized, though again, rank orders of samples are largely unaffected by such effect. For sample storage and long-term preservation, lyophilization (freeze-drying) remains the gold standard for the preservation of biological material, and should be implemented if at all possible [33,72]. Our results confirm this recommendation by showing that freeze-drying also stabilizes urinary suPAR levels. If lyophilization is not an option in the field, then samples should be frozen (regular freezers where there is a power source, solar freezers can be used otherwise).

There was a statistically significant but small decline in concentrations when stored in a cold environment (5–7°C) for up to 2 days prior to freezing. Concentrations were very strongly correlated ($r = 0.98$) with controls, suggesting that such storage is possible without great impacts on suPAR concentrations in either urine (our study) or serum [75]. Nonetheless, rapid processing and freezing of the samples are always preferable. By contrast, storing samples at higher temperatures (more than or equal to 20°C) for two or more days results in changed urinary suPAR concentrations (our study) and serum suPAR levels [75], indicating that longer-term storage at elevated temperatures prior to freezing should be avoided in any case. Multiple freeze–thaw cycles should also be prevented, although our results suggest that it was likely the long, i.e. 24 h, storage at room temperature post-thaw that caused the decrease in values rather than the freezing–thawing–refreezing process itself. We nevertheless recommend keeping the number of freeze–thaw cycles to a minimum by taking care that in cases of power loss of freezers in the field and transport of samples to the laboratory samples do not stay unfrozen for more than a couple of hours. To avoid repeated freeze–thaw situations, researchers may also split the collected urine into several aliquots, a procedure particularly useful if the aim is to measure several compounds from the same sample (e.g. combined analysis of suPAR, uNEO, C-peptide, steroid hormones, creatinine, etc.) [42]. However, note that aliquots of smaller volumes are likely to thaw more quickly during transport.

The specific causes underlying the observed changes in suPAR concentrations in response to the various treatments are not clear, but it is well known that proteins are often vulnerable to conformational change or other types of decomposition (e.g. deglycosylation) under unfavourable conditions [100,101]. If such alterations affect the binding characteristics of the suPAR molecule, changes in measured concentrations as presented here will inevitably be the result. Adding preservatives to enhance protein integrity and stability, such as protease inhibitors or glycols [101], or to reduce bacterial growth using a biocide (e.g. ProClin 200) [74] may help to minimize such 'degradation' effects. Therefore, testing the effectiveness of protein preservatives and stabilizers to hold the binding characteristics of the urinary suPAR molecule steady under the unfavourable conditions potentially encountered in the field could be the next step to improve the validity of suPAR analysis from urine.

In sum, we recommend that contamination of the urine by soil or faeces and exposure to direct sunlight are minimized, that samples are lyophilized if at all possible, but otherwise frozen and then lyophilized once they are shipped back frozen to the laboratory (e.g. on dry ice), with the number of freeze–thaw cycles minimized. With these factors taken into consideration, urinary suPAR seems highly suitable for use in field studies.

# 5. Conclusion

Since uNEO concentrations appear to be more responsive to infection, and suPAR concentrations to inflammation, measurement of both analytes in the same urine sample has great promise for studying different aspects of health status. Although our validation work took place on macaques, urinary suPAR seems likely to be a highly useful biomarker across a much wider taxonomic range. Typically, key molecules involved in immune and other physiological pathways are highly conserved, and in the case of suPAR, the encoding gene PLAUR (NCBI PLAUR Gene ID 5329) is known to be conserved in everything from mice and rats, to dogs, cows, rhesus macaques and chimpanzees [64]. Although there is known alternative splicing that results in different transcriptional variants and isoforms, the fact that both the human and rhesus macaque optimized kits both worked well on our rhesus macaque samples, is consistent with structural conservancy, and indicates that it is likely that the human uPAR assay may work in many other species of primate too, in addition to other species of large mammals.

Although it seems promising that urinary suPAR could have a broad taxonomic application, we strongly encourage more validations along the lines of those we have undertaken here. Biological validations,

involving the demonstration that concentrations rise in response to known inflammation, tissue damage, wounds, infection or specific disease are always important when using a novel non-invasive biomarker in a species in which it has not been measured before [40,43]. Such validation could be experimental, as in the present study, or could take advantage of natural situations in which samples had been, or were, collected before and after observations of wounds and infections [13]. Researchers should also evaluate the potential influence of, for example, age, sex and reproductive state on urinary suPAR levels in the species of interest given that physiological markers are often impacted by these intrinsic factors [102,103]. Furthermore, testing should also consider issues related to field collection and storage, as in the present study. While many of the issues we tested, such as sunlight exposure and freezing, are likely to have very similar effects on the urinary suPAR measurements of other species, faecal bacteria and other microbes may differ substantially between different species, and soil bacteria may differ substantially between soils and environments. The testing of these kinds of potential effects, in particular, may therefore be useful and important. It is also crucial to note that the outcome of stability assessments of a compound measured immunologically as done here are likely specific for the antibody and assay system used. Therefore, whenever a different assay as the one described here is employed to measure urinary suPAR in the species of interest, the type of stability assessments as described here should accompany assay validation. Given the critical importance of measuring health and condition for all kinds of questions in ecology and evolution [1,2], our work suggests that application of urinary suPAR measurement holds promise for field studies of large terrestrial mammals, to address both applied and more academic questions related to animal management, conservation, ecology and evolution.

Ethics. This work adhered to standards as defined by the European Union Council Directive 2010/63/EU on the protection of animals used for scientific purposes. It was also carried out in accordance with the ABS/ASAB guidelines for the ethical treatment of animals, and conformed to the recommendations of the Weatherall report on the use of non-human primates in research. The SIV inoculation experiments (including medical interventions and associated urine sample collections) were approved by the Lower Saxony State Office for Consumer Protection and Food Safety and performed with the project licenses 33.9-42502-04-12/0758-08 (experiment conducted in 2014) and 33.19-42502-04-15/2001 (experiment started in 2016). The SIV and medical intervention experiments were undertaken for other projects, and the present study seeks to take advantage of those experiments to evaluate a non-invasive inflammatory marker.

Data accessibility. Data available from the Dryad Digital Repository: https://doi.org/10.5061/dryad.59zw3r23v [80].

Authors' contributions. J.P.H. and M.H. conceived the study and designed the laboratory experiments. C.S.-H. conducted the SIV experiments and supervised the urine collection. M.H. performed the laboratory experiments and supervised all laboratory analyses. M.H. analysed the data and prepared the figures and tables. J.P.H. and M.H. wrote the manuscript with the help of C.S-H. All authors approved the final draft.

Competing interests. We declare we have no competing interests.

Funding. We received no funding for this study.

Acknowledgements. We thank A. Heistermann for carrying out all enzyme-immunoassays and thank the members of the research group 'Sociality and Health in Primates' (DFG FOR 2136) for stimulating discussions on the general topic. J.H. would like to thank Tara Mandalaywala for sending him a manuscript containing suPAR measurements. We also thank Clare Kimock for help with the references.

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
