## [Reviewer comments · Royal Society Open Science]

Review History

RSOS-191825.R0 (Original submission)

Review form: Reviewer 1

Is the manuscript scientifically sound in its present form?

Yes

Are the interpretations and conclusions justified by the results?

Yes

Is the language acceptable?

Yes

Do you have any ethical concerns with this paper?

No

Have you any concerns about statistical analyses in this paper?

No

Recommendation?

Accept with minor revision (please list in comments)

Comments to the Author(s)

This manuscript presents a validation of urinary suPAR as a general non-invasive measure of inflammation that can be useful for wild animal research. The study is thorough, containing two biological validations of urinary suPAR in response to SIV infection and surgery, and several additional validations to test likely problems in the field (e.g., freeze-thaw, soil contamination, fecal contamination, etc.). My comments here are mostly suggestions for improvement. I detail my comments below:

1. The author's finding of low recovery when standards were spiked with urine (accuracy) is important to bring up in the discussion. Because both parallelism and the second accuracy (with higher recovery) were performed in buffer, I am not concerned with the analytical validation itself. The assay clearly works when urine or lyophilized urine is diluted in buffer. However, the low recovery in pure urine may suggest that a sample running at neat would not be parallel (it would only recover 50% of change) compared to one that is more diluted in buffer. This could have consequences, for instance, if baseline suPAR concentrations are very low in certain animals or even in certain species. I think it is important to talk about this in the discussion.
2. Can the authors comment on the amount of variation in baseline samples and whether there is sufficient sensitivity to detect these differences? Although suPAR is clearly responsive to obviously large infections, it is not clear how well it would reflect smaller differences in baseline immunity. This is important, as many biologists are not just interested in infection-related immune status, but differences in immune biomarkers in non-infected individuals.

Minor comments

Lines 73-75. The wording of this sentence suggests (to me) that the measures mentioned in the paragraph were originally proposed as measures of immunity. Instead, they have been used as proxies for body condition or "stress", of which immune activation could be one culprit. Can the authors make that distinction more clear here?

Lines 220-222 and Lines 243-244. Can the authors comment on the lower detection limit and how that might affect measures of baseline suPAR?

Lines 406-407? Multiple comparison adjustments?

Lines 429. Yes, they are consistent, but that is with one sample (not pooled?). If the interfering substance is variably concentrated across samples, then analyzing a pure (neat) sample might be problematic.

Review form: Reviewer 2

Is the manuscript scientifically sound in its present form?

Yes

Are the interpretations and conclusions justified by the results?

No

Is the language acceptable?

Yes

Do you have any ethical concerns with this paper?

No

Have you any concerns about statistical analyses in this paper?

Yes

Recommendation?

Major revision is needed (please make suggestions in comments)

Comments to the Author(s)

In the manuscript “Urinary suPAR: a non-invasive biomarker of infection and tissue inflammation for use in studies of large free-ranging mammals” Higham et al. present a validation of an ELISA method, suPAR, to examine inflammation and health in a wild primate species. Given that there are few methods available to non-invasively quantify and measure health and immune activation in wild animals, particularly those that cannot be darted, the use of suPAR has the potential to facilitate monitoring health in wild primates. I have a few concerns regarding the applicability of this assay to mammals outside of primates, and also would like to know more about the results of the validation. These concerns are detailed below. Overall I find this validation experiment extremely thorough and informative, covering many of the conditions faced by field biologists.

I find the abstract an adequate representation of the results except that I would be more cautious when stating that suPAR is reasonably robust to most issues associated with field collection, as without a freezer in the field or electricity this method would not be robust, based on the assay results at room temperature. Thus, the authors should be more cautious with this statement. Below I outline my minor concerns.

Introduction:

1. Line 61: perhaps the authors should emphasize also that this is particularly true for any species that are CITES protected – these species typically can only be monitored for health non-invasively which provides another strong point for validated this non-invasive methods
2. Line 69: two other non-invasively methods of energetic condition that could be mentioned here include urea and nitrogen isotopes, both of which can provide information on gluconeogenesis and skeletal muscle wasting (Deschner et al. 2012; Vogel et al, 2012).
3. Line 71-72: Urinary dipstick tests have not been validated for the use on non-human primates or other mammals, other than ketone bodies (Naumenko et al., 2019). Thus, not only is this crude, but I have concerns about whether they are valid for non-human species.
4. Line 106: change to “and in humans, concentrations in urine”
5. Overall, I would like to see more of a justification and review of why suPAR would be better than neopterin for evaluating immune response. Neopterin is relatively stable to freezer thaw cycles (as Higham et al. validated this method as well), and has been validated in a number of non-human primate species. The authors state that suPAR is better for detecting a Th2 response and they predicted that neopterin would not increase in response to inflammation caused by tissue trauma, since it is a more specific marker of cellular immune activation. However, several studies in the human literature have found that neopterin levels rise after surgery and tissue trauma (heart surgery, skin transplants, liver surgery, skin transplants; e.g., Jerin et al. 2005). Thus, while the results of this current study did not find a rise in neopterin levels after surgery, other studies focusing on humans have. This manuscript would benefit from a more detailed review of how suPAR is better and in what situations it would be more appropriate than neopterin. In terms of injury, in many cases when primates have physical injuries these are visible to the observer as well – so how would suPAR be advantageous. One case would be when there is internal injuries that cannot be visualized so this should be emphasized. That said, I am not 100% convinced that external and internal injuries would not be detected with neopterin.

Methods:

6. In general, it appears that male and female samples were pooled for all analyses. While the sample size is very small for each experiment (n=6-7 individuals), would it be possible to test for the effect of sex? This would only be important for the 2014 experiments as in the 2017 experiments the subjects were all males.

7. I would like to see a breakdown of the number of samples from each individual in a table format for each experiment. For example, I am confused about the sample sizes that went into each analysis. For example, for the HIV experiment there are 83 samples across 7 individuals so this is about 11 samples per individual split to before and after the SIV infection? So this is about 5 samples for the control baseline and 5 samples for after the infection? Prior to virus inoculation there were 21 samples (3 samples taken 1-7 weeks prior to inoculation – I assume this is 3 samples per individual for 7 weeks?) before and after, so this should be $3 \times 4 \times 7 = 84$ samples. This should be then 105 samples - I would like to see a breakdown of the number of samples taken from each individual in each time period for the SIV experiment in table form to clarify the sampling methods used for this study. In addition, the authors should add sample size to each treatment in Table 2.
8. Given that the recovery was so low (45.5%), it would be important to present the recovery results for each dilution of assay standard added. Was the low recovery consistent across all dilutions of the spiked samples or was there variation among the high and low spiked samples in terms of recovery (this is said in line 428 but no data are provided) – please provide in a supplement or table? Typically no matrix interference is considered between 80-120% recovery – thus 45.5% is very low and high matrix interference which really questions the validity of this assay.
9. Lines 244-256: Why even include the samples that fell below the standard curve? If the samples are excluded, does it affect the results? Setting them equal to the minimum detection limit is (you say maximum value possible which does not really make sense as that would be the highest standard in the curve) arbitrary unless a difference of 0-15 would not make a difference. Running the analysis with and without the samples and examining explained variance would let you know how much those samples are influencing the data set
10. For the assay, it is not clear why the authors are reporting the intra-assay CV that is reported by the kit manufacturer. Why not report the intra-assay CV from the high and low QCs as the authors do with inter-assay CV?
11. I wonder why the authors did not examine if there is a correlation between uNeo and supAR as a matched samples test? This would be interesting. While proper analyses would control for individual ID, since this is a validation and samples sizes are too small for this type of analysis this cannot be done. However, I still think it is useful to see if the matched samples for uNEO and suPAR correlate?
12. Line 258: same as comment 10 for uNEO
13. Line 254-261: It is not clear why suPAR and uNEO are corrected for Cr instead of SG when in the results and discussion the authors say SG is preferred method due to stability and muscle mass issues. I would like to see all of the results rerun with SG correction - are the results the same? This is particularly important given that males and female are being used in the same analysis in some cases. It is also confusing because then in Line 320-331 they authors state both CR and SG were used – this needs some clarification as it is very confusing what was actually used to standardize the samples. This needs some reworking
14. Line 273: Is this a matched test?
15. Line 324: was Cr and SG measured on fresh samples or both on the samples that had been frozen?
16. Line 332: change to “As CR and SG are also widely.....”
17. Line 372: remove “In order to” and just say “To evaluate....”
18. Line 401: Did the authors check for sex differences before pooling the data?
19. Line 404: Were these analyses done as a matched tests? If done in R this is quite simple.
20. Was a power analyses done?

Results

21. Line 420-421: I am a little confused that a t-stat is reported here for a samples size that is less than 25? If $df=8$, samples size is 9 and thus this should be reported as a non-parametric statistic.
22. Line 445: With regards to the uNEO result here, it looks that the elevation at the end of the 29 days is only about 100 ng/ml above the pre – the question is, is this rise biologically relevant?

23. Lines 481-483: This statement would apply to all analyses. One would expect a tight correlation if only the mean is changing consistently across samples but the variance does not change with treatment? Thus, I am not sure what the point of this is unless the authors are suggesting that a correction factor can be used because the mean observation is lower in the experimental treatment but the variance does not change? Does the variance change or just the means? The authors show that the values are correlated but what does this really tell us?
24. Lines 510, 513, 529, 532, 549, 560: The results are often reported that the rank order of samples between controls and experiments remain “basically” unchanged or “largely” unchanged. I see this vague language as problematic and would rather see a percentage of changes in rank order reported instead of using this language.
25. Line 548: same comment as number 23 – with one freezer thaw cycle the values decline but are correlated with controls – suggesting a consistent decline in mean values and similar variance. Can the authors confirm this and what is the biological meaning of reporting this – are the authors suggesting a correction equation could be applied if thawed vs non-previously thawed samples are used in the same analysis?
26. Line 576: I am still convinced of the applicability of this assay in general to free ranging mammals. I am convinced that this human kit may be applied potentially to come other primate species if the samples have been collected and frozen and not thawed. For macaques and other primates maybe, but I am not convinced this would work for other mammals so this language should be removed.
27. 584-585: The language used here should be more cautious. I think it is important to note that this method can be used in only certain field conditions where there is reliable electricity for a freezer.
28. Lines 620-630: I would like a deeper discussion of why suPAR is a superior marker to uNEO and under what conditions one would want to use suPAR instead of uNEO in the field. Given that uNEO is more stable to field conditions, it would seem that it may be advantageous.
29. Line 660 and 666: I would like to see this “largely” quantified.
30. Lines 672-673: It seems like there was actually an impact – 2 days after there was a significant difference from the controls unless I am reading the data incorrectly?
31. Line 683-687: While smaller aliquots may prevent multiple freezer thaw cycles, they also are more likely to thaw completely in transit if dry ice or liquid nitrogen is not used. When samples are aliquoted in 2ml tubes, they are much less likely to thaw compared to those that are aliquoted in say 200ul samples. Thus, there are costs and benefits to each method and this should be noted.
32. Line 703 – except for the very low recovery with macaque urine. What was the recovery with human urine?
33. Line 739: I think this is a bit of an overreach given that the human kit revealed very low recovery in a non-human primate species – do the authors really think this kit holds great promise for field studies of large terrestrial mammals (especially given the soil results?).

Decision letter (RSOS-191825.R0)

04-Dec-2019

Dear Dr Higham,

On behalf of the Editors, I am pleased to inform you that your Manuscript RSOS-191825 entitled "Urinary suPAR: a non-invasive biomarker of infection and tissue inflammation for use in studies of large free-ranging mammals" has been accepted for publication in Royal Society Open Science subject to minor revision in accordance with the referee suggestions. Please find the referees' comments at the end of this email.

The reviewers and handling editors have recommended publication, but also suggest some minor

revisions to your manuscript. Therefore, I invite you to respond to the comments and revise your manuscript.

- Ethics statement

- Data accessibility

If you wish to submit your supporting data or code to Dryad (<http://datadryad.org/>), or modify your current submission to dryad, please use the following link:
<http://datadryad.org/submit?journalID=RSOS&manu=RSOS-191825>

- Competing interests

- Authors' contributions

- Acknowledgements

- Funding statement

Please ensure you have prepared your revision in accordance with the guidance at <https://royalsociety.org/journals/authors/author-guidelines/> -- please note that we cannot publish your manuscript without the end statements. We have included a screenshot example of

the end statements for reference. If you feel that a given heading is not relevant to your paper, please nevertheless include the heading and explicitly state that it is not relevant to your work.

Because the schedule for publication is very tight, it is a condition of publication that you submit the revised version of your manuscript before 3 January 2020. Please note that the revision deadline will expire at 00.00am on this date. If you do not think you will be able to meet this date please let us know immediately.

If your manuscript is newly submitted and subsequently accepted for publication, you will be asked to pay the article processing charge, unless you request a waiver and this is approved by Royal Society Publishing. You can find out more about the charges at <https://royalsocietypublishing.org/rsos/charges>. Should you have any queries, please contact openscience@royalsociety.org.

Best regards,

on behalf of Dr Alexander Ophir (Associate Editor) and Kevin Padian (Subject Editor)
openscience@royalsociety.org

Associate Editor Comments to Author (Dr Alexander Ophir):

Dear Dr. Higham,

Your manuscript has now been seen by two expert referees whose reports are at the end of this email. Your study addresses offers an interesting way to assess a non-invasively collected biomarker of health that should generate some general interest. Although reviewer 1 found your validations very compelling and offered only minor suggestions, reviewer 2 was less convinced that your measure will generalize beyond primates. This reviewer has provided a mix of very minor and somewhat more substantial comments that should serve as very useful feedback, and I believe you will be able to adequately address these in a revision. Overall, there is indeed a need to increase the number of validated non-invasive methods to collect biomarkers of health, and I think your report will represent an additional useful tool toward this end.

Best
Alex Ophir
Associate Editor, RSOS

Reviewer comments to Author:

Reviewer: 1
Comments to the Author(s)

This manuscript presents a validation of urinary suPAR as a general non-invasive measure of inflammation that can be useful for wild animal research. The study is thorough, containing two biological validations of urinary suPAR in response to SIV infection and surgery, and several additional validations to test likely problems in the field (e.g., freeze-thaw, soil contamination, fecal contamination, etc.). My comments here are mostly suggestions for improvement. I detail my comments below:

1. The author's finding of low recovery when standards were spiked with urine (accuracy) is important to bring up in the discussion. Because both parallelism and the second accuracy (with

higher recovery) were performed in buffer, I am not concerned with the analytical validation itself. The assay clearly works when urine or lyophilized urine is diluted in buffer. However, the low recovery in pure urine may suggest that a sample running at neat would not be parallel (it would only recover 50% of change) compared to one that is more diluted in buffer. This could have consequences, for instance, if baseline suPAR concentrations are very low in certain animals or even in certain species. I think it is important to talk about this in the discussion.

2. Can the authors comment on the amount of variation in baseline samples and whether there is sufficient sensitivity to detect these differences? Although suPAR is clearly responsive to obviously large infections, it is not clear how well it would reflect smaller differences in baseline immunity. This is important, as many biologists are not just interested in infection-related immune status, but differences in immune biomarkers in non-infected individuals.

Minor comments

Lines 73-75. The wording of this sentence suggests (to me) that the measures mentioned in the paragraph were originally proposed as measures of immunity. Instead, they have been used as proxies for body condition or "stress", of which immune activation could be one culprit. Can the authors make that distinction more clear here?

Lines 220-222 and Lines 243-244. Can the authors comment on the lower detection limit and how that might affect measures of baseline suPAR?

Lines 406-407? Multiple comparison adjustments?

Lines 429. Yes, they are consistent, but that is with one sample (not pooled?). If the interfering substance is variably concentrated across samples, then analyzing a pure (neat) sample might be problematic.

Reviewer: 2

Comments to the Author(s)

In the manuscript "Urinary suPAR: a non-invasive biomarker of infection and tissue inflammation for use in studies of large free-ranging mammals" Higham et al. present a validation of an ELISA method, suPAR, to examine inflammation and health in a wild primate species. Given that there are few methods available to non-invasively quantify and measure health and immune activation in wild animals, particularly those that cannot be darted, the use of suPAR has the potential to facilitate monitoring health in wild primates. I have a few concerns regarding the applicability of this assay to mammals outside of primates, and also would like to know more about the results of the validation. These concerns are detailed below. Overall I find this validation experiment extremely thorough and informative, covering many of the conditions faced by field biologists.

I find the abstract an adequate representation of the results except that I would be more cautious when stating that suPAR is reasonably robust to most issues associated with field collection, as without a freezer in the field or electricity this method would not be robust, based on the assay results at room temperature. Thus, the authors should be more cautious with this statement. Below I outline my minor concerns.

Introduction:

1. Line 61: perhaps the authors should emphasize also that this is particularly true for any species that are CITES protected - these species typically can only be monitored for health non-invasively which provides another strong point for validated this non-invasive methods
2. Line 69: two other non-invasively methods of energetic condition that could be mentioned here include urea and nitrogen isotopes, both of which can provide information on gluconeogenesis and skeletal muscle wasting (Deschner et al. 2012; Vogel et al, 2012).

3. Line 71-72: Urinary dipstick tests have not been validated for the use on non-human primates or other mammals, other than ketone bodies (Naumenko et al., 2019). Thus, not only is this crude, but I have concerns about whether they are valid for non-human species.

4. Line 106: change to “and in humans, concentrations in urine

5. Overall, I would like to see more of a justification and review of why suPAR would be better than neopterin for evaluating immune response. Neopterin is relatively stable to freezer thaw cycles (as Higham et al. validated this method as well), and has been validated in a number of non-human primate species. The authors state that suPAR is better for detecting a Th2 response and they predicted that neopterin would not increase in response to inflammation caused by tissue trauma, since it is a more specific marker of cellular immune activation. However, several studies in the human literature have found that neopterin levels rise after surgery and tissue trauma (heart surgery, skin transplants, liver surgery, skin transplants; e.g., Jerin et al. 2005). Thus, while the results of this current study did not find a rise in neopterin levels after surgery, other studies focusing on humans have. This manuscript would benefit from a more detailed review of how suPAR is better and in what situations it would be more appropriate than neopterin. In terms of injury, in many cases when primates have physical injuries these are visible to the observer as well – so how would suPAR be advantageous. One case would be when there is internal injuries that cannot be visualized so this should be emphasized. That said, I am not 100% convinced that external and internal injuries would not be detected with neopterin.

Methods:

6. In general, it appears that male and female samples were pooled for all analyses. While the sample size is very small for each experiment (n=6-7 individuals), would it be possible to test for the effect of sex? This would only be important for the 2014 experiments as in the 2017 experiments the subjects were all males.

7. I would like to see a breakdown of the number of samples from each individual in a table format for each experiment. For example, I am confused about the sample sizes that went into each analysis. For example, for the HIV experiment there are 83 samples across 7 individuals so this is about 11 samples per individual split to before and after the SIV infection? So this is about 5 samples for the control baseline and 5 samples for after the infection? Prior to virus inoculation there were 21 samples (3 samples taken 1-7 weeks prior to inoculation – I assume this is 3 samples per individual for 7 weeks?) before and after, so this should be $3 \times 4 \times 7 = 84$ samples. This should be then 105 samples - I would like to see a breakdown of the number of samples taken from each individual in each time period for the SIV experiment in table form to clarify the sampling methods used for this study. In addition, the authors should add sample size to each treatment in Table 2.

8. Given that the recovery was so low (45.5%), it would be important to present the recovery results for each dilution of assay standard added. Was the low recovery consistent across all dilutions of the spiked samples or was there variation among the high and low spiked samples in terms of recovery (this is said in line 428 but no data are provided) – please provide in a supplement or table? Typically no matrix interference is considered between 80-120% recovery – thus 45.5% is very low and high matrix interference which really questions the validity of this assay.

9. Lines 244-256: Why even include the samples that fell below the standard curve? If the samples are excluded, does it affect the results? Setting them equal to the minimum detection limit is (you say maximum value possible which does not really make sense as that would be the highest standard in the curve) arbitrary unless a difference of 0-15 would not make a difference. Running the analysis with and without the samples and examining explained variance would let you know how much those samples are influencing the data set

10. For the assay, it is not clear why the authors are reporting the intra-assay CV that is reported by the kit manufacturer. Why not report the intra-assay CV from the high and low QCs as the authors do with inter-assay CV?

11. I wonder why the authors did not examine if there is a correlation between uNeo and supAR as a matched samples test? This would be interesting. While proper analyses would control for individual ID, since this is a validation and samples sizes are too small for this type of analysis

this cannot be done. However, I still think it is useful to see if the matched samples for uNEO and suPAR correlate?

12. Line 258: same as comment 10 for uNEO

13. Line 254-261: It is not clear why suPAR and uNEO are corrected for Cr instead of SG when in the results and discussion the authors say SG is preferred method due to stability and muscle mass issues. I would like to see all of the results rerun with SG correction - are the results the same? This is particularly important given that males and female are being used in the same analysis in some cases. It is also confusing because then in Line 320-331 they authors state both CR and SG were used - this needs some clarification as it is very confusing what was actually used to standardize the samples. This needs some reworking

14. Line 273: Is this a matched test?

15. Line 324: was Cr and SG measured on fresh samples or both on the samples that had been frozen?

16. Line 332: change to "As CR and SG are also widely....."

17. Line 372: remove "In order to" and just say "To evaluate...."

18. Line 401: Did the authors check for sex differences before pooling the data?

19. Line 404: Were these analyses done as a matched tests? If done in R this is quite simple.

20. Was a power analyses done?

Results

21. Line 420-421: I am a little confused that a t-stat is reported here for a samples size that is less than 25? If $df=8$, samples size is 9 and thus this should be reported as a non-parametric statistic.

22. Line 445: With regards to the uNEO result here, it looks that the elevation at the end of the 29 days is only about 100 ng/ml above the pre - the question is, is this rise biologically relevant?

23. Lines 481-483: This statement would apply to all analyses. One would expect a tight correlation if only the mean is changing consistently across samples but the variance does not change with treatment? Thus, I am not sure what the point of this is unless the authors are suggesting that a correction factor can be used because the mean observation is lower in the experimental treatment but the variance does not change? Does the variance change or just the means? The authors show that the values are correlated but what does this really tell us?

24. Lines 510, 513, 529, 532, 549, 560: The results are often reported that the rank order of samples between controls and experiments remain "basically" unchanged or "largely" unchanged. I see this vague language as problematic and would rather see a percentage of changes in rank order reported instead of using this language.

25. Line 548: same comment as number 23 - with one freezer thaw cycle the values decline but are correlated with controls - suggesting a consistent decline in mean values and similar variance. Can the authors confirm this and what is the biological meaning of reporting this - are the authors suggesting a correction equation could be applied if thawed vs non-previously thawed samples are used in the same analysis?

26. Line 576: I am still convinced of the applicability of this assay in general to free ranging mammals. I am convinced that this human kit may be applied potentially to come other primate species if the samples have been collected and frozen and not thawed. For macaques and other primates maybe, but I am not convinced this would work for other mammals so this language should be removed.

27. 584-585: The language used here should be more cautious. I think it is important to note that this method can be used in only certain field conditions where there is reliable electricity for a freezer.

28. Lines 620-630: I would like a deeper discussion of why suPAR is a superior marker to uNEO and under what conditions one would want to use suPAR instead of uNEO in the field. Given that uNEO is more stable to field conditions, it would seem that it may be advantageous.

29. Line 660 and 666: I would like to see this "largely" quantified.

30. Lines 672-673: It seems like there was actually an impact - 2 days after there was a significant difference from the controls unless I am reading the data incorrectly?

31. Line 683-687: While smaller aliquots may prevent multiple freezer thaw cycles, they also are more likely to thaw completely in transit if dry ice or liquid nitrogen is not used. When samples are aliquoted in 2ml tubes, they are much less likely to thaw compared to those that are aliquoted in say 200ul samples. Thus, there are costs and benefits to each method and this should be noted.

32. Line 703 – except for the very low recovery with macaque urine. What was the recovery with human urine?

33. Line 739: I think this is a bit of an overreach given that the human kit revealed very low recovery in a non-human primate species – do the authors really think this kit holds great promise for field studies of large terrestrial mammals (especially given the soil results?).

Author's Response to Decision Letter for (RSOS-191825.R0)

See Appendix A.

Decision letter (RSOS-191825.R1)

17-Jan-2020

Dear Dr Higham,

It is a pleasure to accept your manuscript entitled "Urinary suPAR: a non-invasive biomarker of infection and tissue inflammation for use in studies of large free-ranging mammals" in its current form for publication in Royal Society Open Science. The comments of the reviewer(s) who reviewed your manuscript are included at the foot of this letter.

Kind regards,
Andrew Dunn
Royal Society Open Science Editorial Office

on behalf of Dr Alexander Ophir (Associate Editor) and Kevin Padian (Subject Editor)
openscience@royalsociety.org

Appendix A

Dear Editors and Reviewers,

Please find attached a new version of RSOS-191825, which has been revised in response to editor and reviewer comments. We are extremely grateful to the Editor and Reviewers for their constructive feedback, which has improved the manuscript substantially. Below we provide detailed responses to each comment, giving line numbers that correspond to the new version.

Dear Dr. Higham,

Your manuscript has now been seen by two expert referees whose reports are at the end of this email. Your study addresses offers an interesting way to assess a non-invasively collected biomarker of health that should generate some general interest. Although reviewer 1 found your validations very compelling and offered only minor suggestions, reviewer 2 was less convinced that your measure will generalize beyond primates. This reviewer has provided a mix of very minor and somewhat more substantial comments that should serve as very useful feedback, and I believe you will be able to adequately address these in a revision. Overall, there is indeed a need to increase the number of validated non-invasive methods to collect biomarkers of health, and I think your report will represent an additional useful tool toward this end.

Best
Alex Ophir
Associate Editor, RSOS

We thank the Editor for his positive recommendation of provisional acceptance, and for his constructive feedback on the manuscript.

Reviewer comments to Author:

Reviewer: 1
Comments to the Author(s)

This manuscript presents a validation of urinary suPAR as a general non-invasive measure of inflammation that can be useful for wild animal research. The study is thorough, containing two biological validations of urinary suPAR in response to SIV infection and surgery, and several additional validations to test likely problems in the field (e.g., freeze-thaw, soil contamination, fecal contamination, etc.). My comments here are mostly suggestions for improvement. I detail my comments below:

We thank the Reviewer for their constructive and helpful comments.

1. The author's finding of low recovery when standards were spiked with urine (accuracy) is important to bring up in the discussion. Because both parallelism and the second accuracy (with higher recovery) were performed in buffer, I am not concerned with the analytical validation itself. The assay clearly works when urine or lyophilized urine is diluted in buffer. However, the low recovery in pure urine may

suggest that a sample running at neat would not be parallel (it would only recover 50% of change) compared to one that is more diluted in buffer. This could have consequences, for instance, if baseline suPAR concentrations are very low in certain animals or even in certain species. I think it is important to talk about this in the discussion.

We thank the Reviewer for these thoughtful comments. For the accuracy test, we actually diluted the pooled urine 1:6 with assay buffer to lower the endogenous suPAR concentrations prior to spiking the sample with suPAR standards. The suPAR standards used for spiking were also diluted in assay buffer. As such, the urine accuracy test was carried out with samples diluted in assay buffer and was not done in pure urine as assumed by the reviewer. This misunderstanding is our fault – we were unclear about this in the manuscript, for which we apologize. We have now added text to the revised manuscript to make this clear (Lines 231-239).

As the reviewer notes, the fact that the samples dilute perfectly in parallel to each other and to dilutions of the standard, indicate that the degree of interference is not dependent on the amount of urine in the sample and that relative differences in suPAR levels between samples are unaffected by the low overall recovery.

2. Can the authors comment on the amount of variation in baseline samples and whether there is sufficient sensitivity to detect these differences? Although suPAR is clearly responsive to obviously large infections, it is not clear how well it would reflect smaller differences in baseline immunity. This is important, as many biologists are not just interested in infection-related immune status, but differences in immune biomarkers in non-infected individuals.

We thank the reviewer for this comment. We would like to emphasize that suPAR is most responsive to inflammation rather than infection. For example, it is less responsive than neopterin to SIV treatment, but responds strongly to tissue damage associated with surgery and colonoscopy. We therefore have edited the Discussion to be clearer that for studies interested in monitoring general health we recommend measurement of both alongside each other – NEO for immune activation, and suPAR for both infection-related and tissue-damage-related inflammation (Lines 766-768).

Regarding baseline variation: the variation in individual mean pre-treatment baseline values ranged between 0.33 and 0.94 ng/mg Cr in animals from the 2014 study, and between 0.27 and 1.19 ng/mg Cr in animals from the 2017 study, representing a 3-5-fold difference in baseline values between individuals. As such, samples show a good deal of baseline variation between different individuals. We have now added this information to the manuscript to give readers a clearer idea of the potential of suPAR measurement for tracking baseline variation (Lines 442-448; Lines 467-470). We also now discuss the issue in the revised submission (Lines 656-659)

Minor comments

Lines 73-75. The wording of this sentence suggests (to me) that the measures mentioned in the paragraph were originally proposed as measures of immunity. Instead, they have been used as proxies for body condition or "stress", of which immune activation could be one culprit. Can the authors make that distinction more clear here?

We have clarified the point here, revising the sentence to say that *“Though useful in many aspects, these measures have usually been proposed for measuring other aspects of physical condition, such as energetic status, and either do not reflect the infectious status or degree of immune activation or inflammatory status of an individual at all, or do so only very crudely and/or indirectly.”* (Lines 75-78).

Lines 220-222 and Lines 243-244. Can the authors comment on the lower detection limit and how that might affect measures of baseline suPAR?

Of the baseline samples of the two studies, only 11.1% (2014) and 6.3% (2017) of the samples were below assay sensitivity with the 1:5 dilution used. In other words, about 90% of baseline samples were within the detection limit of the assay under the conditions described. It is likely that a higher proportion of samples would be detectable if samples were diluted less (e.g. 1:2 or 1:3). As such, we envisage that the suPAR assay described here would be well able to detect small-scale differences in baseline immunity in non-infected animals. We have added this information to the manuscript (Lines 245-249).

Lines 406-407? Multiple comparison adjustments?

Since we do not repeated testing within any one model of set of dependent variables, but undertake tests of separate conditions, we do not believe that this is necessary here.

Lines 429. Yes, they are consistent, but that is with one sample (not pooled?). If the interfering substance is variably concentrated across samples, then analyzing a pure (neat) sample might be problematic.

The urine used for the accuracy test was a male-female pooled sample. The results of our parallelism test indicate that the amount/proportion of urine in the sample (and thus the amount of the substance that is interfering in the assay matrix and reducing the recovery rate) does not affect the strength of the interference. Samples diluted perfectly parallel to the standard curve. If variable concentration in the interfering substance altered the strength of the interference, then highly diluted samples would be less affected than low diluted samples - this is not the case. It thus seems that the interfering effect impacts suPAR levels in terms of leading to overall underestimated concentrations, but that it does not affect relative levels between samples/individuals. We have now clarified this in the Discussion (Lines 619-627).

Reviewer: 2

Comments to the Author(s)

In the manuscript “Urinary suPAR: a non-invasive biomarker of infection and tissue inflammation for use in studies of large free-ranging mammals” Higham et al. present a validation of an ELISA method, suPAR, to examine inflammation and health in a wild primate species. Given that there are few methods available to non-invasively quantify and measure health and immune activation in wild animals, particularly those that cannot be darted, the use of suPAR has the potential to facilitate monitoring health in wild primates. I have a few concerns regarding the applicability of this assay to mammals outside of primates, and also would like to know more about the results of the validation. These

concerns are detailed below. Overall I find this validation experiment extremely thorough and informative, covering many of the conditions faced by field biologists.

We thank the Reviewer for their constructive and helpful comments.

I find the abstract an adequate representation of the results except that I would be more cautious when stating that suPAR is reasonably robust to most issues associated with field collection, as without a freezer in the field or electricity this method would not be robust, based on the assay results at room temperature. Thus, the authors should be more cautious with this statement. Below I outline my minor concerns.

We have tweaked the abstract to include this important caveat – we now state that the methods seem likely to be robust as long as fieldworkers are able to store samples in a freezer (Lines 35-36).

Introduction:

1. Line 61: perhaps the authors should emphasize also that this is particularly true for any species that are CITES protected – these species typically can only be monitored for health non-invasively which provides another strong point for validated this non-invasive methods

This point has been added (Lines 62-63).

2. Line 69: two other non-invasively methods of energetic condition that could be mentioned here include urea and nitrogen isotopes, both of which can provide information on gluconeogenesis and skeletal muscle wasting (Deschner et al. 2012; Vogel et al, 2012).

This has been added, as suggested (Lines 72-73).

3. Line 71-72: Urinary dipstick tests have not been validated for the use on non-human primates or other mammals, other than ketone bodies (Naumenko et al., 2019). Thus, not only is this crude, but I have concerns about whether they are valid for non-human species.

We agree, and have clarified in the revised manuscript that to our knowledge such approaches have not been biologically validated (Lines 74-75).

4. Line 106: change to “and in humans, concentrations in urine”

This change has been made (Line 109).

5. Overall, I would like to see more of a justification and review of why suPAR would be better than neopterin for evaluating immune response. Neopterin is relatively stable to freezer thaw cycles (as Higham et al. validated this method as well), and has been validated in a number of non-human primate species. The authors state that suPAR is better for detecting a Th2 response and they predicted that neopterin would not increase in response to inflammation caused by tissue trauma, since it is a more specific marker of cellular immune activation. However, several studies in the human literature have

found that neopterin levels rise after surgery and tissue trauma (heart surgery, skin transplants, liver surgery, skin transplants; e.g., Jerin et al. 2005). Thus, while the results of this current study did not find a rise in neopterin levels after surgery, other studies focusing on humans have. This manuscript would benefit from a more detailed review of how suPAR is better and in what situations it would be more appropriate than neopterin. In terms of injury, in many cases when primates have physical injuries these are visible to the observer as well – so how would suPAR be advantageous. One case would be when there is internal injuries that cannot be visualized so this should be emphasized. That said, I am not 100% convinced that external and internal injuries would not be detected with neopterin.

As stated above in our response to Reviewer 1, we do believe that there are good reasons to think that NEO is likely to be more responsive to infection, and is more specifically a marker of immunity related to intracellular immune activation as part of the TH1 immune response, while suPAR is related to inflammation (including infection-related inflammation). They are also backed by our data in the present study. While it's true that some studies of surgery in humans have measured neopterin increases, it is also true that most surgeries in hospitals involve some level of infection, and typically antibiotics are taken following surgery for that reason. We advocate that both approaches have their uses, and that both could be measured from the same samples if the goal is to measure health generally. We now discuss these differences more clearly (Lines 605-610) and advocate this explicitly (Lines 766-768).

Methods:

6. In general, it appears that male and female samples were pooled for all analyses. While the sample size is very small for each experiment (n=6-7 individuals), would it be possible to test for the effect of sex? This would only be important for the 2014 experiments as in the 2017 experiments the subjects were all males.

We don't believe that the sample sizes are sufficient for a sex analysis, given just 3 males and 3 females. Nonetheless it may be useful to readers to have the breakdown by sex. Pre-treatment baseline concentration were: males, 0.57 +/- 0.32 ng/mg Cr; females, 0.42 +/- 0.14 ng/mg Cr. In terms of strengths of response to surgery, median responses were: males, 10.7-fold increase; females, 7.9-fold increase). From this, it certainly seems that there were no marked sex-specific effects. We have added this information to the revised manuscript (Lines 442-446).

7. I would like to see a breakdown of the number of samples from each individual in a table format for each experiment. For example, I am confused about the sample sizes that went into each analysis. For example, for the HIV experiment there are 83 samples across 7 individuals so this is about 11 samples per individual split to before and after the SIV infection? So this is about 5 samples for the control baseline and 5 samples for after the infection? Prior to virus inoculation there were 21 samples (3 samples taken 1-7 weeks prior to inoculation – I assume this is 3 samples per individual for 7 weeks?) before and after, so this should be $3 \times 4 \times 7 = 84$ samples. This should be then 105 samples - I would like to see a breakdown of the number of samples taken from each individual in each time period for the SIV experiment in table form to clarify the sampling methods used for this study. In addition, the authors should add sample size to each treatment in Table 2.

We have added more details to the text to clarify the sample sizes (Lines 191-207). The sample sizes for the different treatments tested as part of our field validation study were always n=8. We have now indicated this in the table legend.

8. Given that the recovery was so low (45.5%), it would be important to present the recovery results for each dilution of assay standard added. Was the low recovery consistent across all dilutions of the spiked samples or was there variation among the high and low spiked samples in terms of recovery (this is said in line 428 but no data are provided) – please provide in a supplement or table? Typically no matrix interference is considered between 80-120% recovery – thus 45.5% is very low and high matrix interference which really questions the validity of this assay.

The low recovery is indeed perplexing to us. However, it is highly consistent – as stated in the manuscript, the standard deviation of recovery measurements was just 4.7% around the 45.5% recovery – just 10% variation. As such, while only around half of the suPAR is measured, it is reliably and consistently half. As stated above, the parallelism test indicates that changes in the concentration of urine do not affect measurements, such that the measurements are analytically valid. Finally, the biological validation itself clearly shows that measurements are reliable and informative in detecting inflammation. We have added these clarifying points to the Discussion (Lines 619-627).

9. Lines 244-256: Why even include the samples that fell below the standard curve? If the samples are excluded, does it affect the results? Setting them equal to the minimum detection limit is (you say maximum value possible which does not really make sense as that would be the highest standard in the curve) arbitrary unless a difference of 0-15 would not make a difference. Running the analysis with and without the samples and examining explained variance would let you know how much those samples are influencing the data set

As stated above, only a few samples were below detection limit. If sample values are below minimum assay sensitivity, then by definition the maximum possible value they could actually have (if it were possible to measure them) is just below that minimum sensitivity value. Setting sample values as equal to that value is underestimating the variance in the dataset, because in reality many of those samples are probably even lower than this assumed value. We believe that this is a conservative approach. We have clarified this in the text. We also confirmed that excluding these values does not affect the results, and have also added this information to the text (Lines 249-253).

10. For the assay, it is not clear why the authors are reporting the intra-assay CV that is reported by the kit manufacturer. Why not report the intra-assay CV from the high and low QCs as the authors do with inter-assay CV

We have now included mean intra-assay QC values as calculated from the high and low QCs run on each plate (Lines 255-256).

11. I wonder why the authors did not examine if there is a correlation between uNeo and suPAR as a matched samples test? This would be interesting. While proper analyses would control for individual ID, since this is a validation and samples sizes are too small for this type of analysis this cannot be done. However, I still think it is useful to see if the matched samples for uNEO and suPAR correlate?

As the Reviewer notes, we would need to control for Individual ID to undertake such an analysis correctly, but we do not have sufficient sample sizes. We did run a Pearson correlation and it was $r=0.64$ for the samples collected in 2014 during the SIV infection study and $r=0.51$ for the samples

collected 2017 for the surgery study - both values are highly significant. We have not added this to the methods or results, since the approach is incorrect. However, we now mention it in the Discussion, while acknowledging that the statistical approach is itself pseudo-replicated and urging caution, since we agree with the Reviewer that it is potentially of interest to readers (Lines 661-668).

12. Line 258: same as comment 10 for uNEO

We have now included intra-assay QC values as calculated from high and low QCs run on each plate (Lines 264-265).

13. Line 254-261: It is not clear why suPAR and uNEO are corrected for Cr instead of SG when in the results and discussion the authors say SG is preferred method due to stability and muscle mass issues. I would like to see all of the results rerun with SG correction - are the results the same? This is particularly important given that males and female are being used in the same analysis in some cases. It is also confusing because then in Line 320-331 they authors state both CR and SG were used – this needs some clarification as it is very confusing what was actually used to standardize the samples. This needs some reworking.

In our experience analyte values indexed for Cr and for SG are almost identical and show the same pattern. Nonetheless, the reviewer is correct that our results in the field validation experiments show that SG measurements are more reliable for urine dilution correction under unfavourable conditions, such as contamination of urine with feces or storage samples at ambient temperature or when animal diet may change. Therefore, we recommend in the MS the use of SG for indexing urinary analytes when samples are collected under uncontrollable conditions from wild animals. The situation is different for captive-housed animals where such unfavourable conditions do not occur, or where they can easily be avoided, and where animals live in a controlled and consistent environment. Here, creatinine is similarly reliable as SG for indexing purposes. To confirm this for the present study, we thawed 28 of the urine samples from the present study, and measured them for SG, and plotted the correlation between the Cr values and the SG values. The r of this correlation was 0.97. As such, we can be sure that values indexed for SG rather than Cr would produce exactly the same results. We have added this information to the manuscript (Lines 268-271). We are reluctant to thaw all of the remaining samples, as they are valuable and can be used for other validations, which could be affected by the additional freeze-thaw cycle.

14. Line 273: Is this a matched test?

This is a Spearman's rank test. To our knowledge this is never a matched test as it uses ordinal ranking of the two sample sets.

15. Line 324: was Cr and SG measured on fresh samples or both on the samples that had been frozen?

Both Cr and SG were initially measured in fresh samples and afterwards in all treatment samples (including controls) to assess the effect of treatment on Cr and SG values. However, samples were not measured for SG for the biological validation (but see above). These clarifications have been added to the manuscript (Lines 328-329).

16. Line 332: change to "As CR and SG are also widely....."

This has been changed as suggested (Line 342).

17. Line 372: remove “In order to” and just say “To evaluate....”

This has been changed as suggested (Line 382).

18. Line 401: Did the authors check for sex differences before pooling the data?

The samples used for the field validation experiments were all from males.

19. Line 404: Were these analyses done as a matched tests? If done in R this is quite simple.

To our knowledge, the Wilcoxon Signed Ranks test is always a matched-sample test, and is the non-parametric equivalent of a paired t-test.

20. Was a power analyses done?

We did not undertake a power analysis for this exploratory study. To our knowledge, there was no available data that would suggest what potential effect sizes might look like.

Results

21. Line 420-421: I am a little confused that a t-stat is reported here for a samples size that is less than 25? If $df=8$, samples size is 9 and thus this should be reported as a non-parametric statistic.

This is not a t-test, but a Tukey test, which also calculates a t statistic. We have added this clarification to the manuscript (Lines 281 and 430).

22. Line 445: With regards to the uNEO result here, it looks that the elevation at the end of the 29 days is only about 100 ng/ml above the pre – the question is, is this rise biologically relevant?

The graph that the reviewer refers to has a y-axis that shows percentage values rather than absolute values. The data point on day 29 post-SIV inoculation shows that uNEO levels are still elevated by around 200% compared to pre-treatment baseline values. This is a substantial elevation which is probably biologically meaningful as it indicates a persistent cellular immune activation in SIV-infected animals.

23. Lines 481-483: This statement would apply to all analyses. One would expect a tight correlation if only the mean is changing consistently across samples but the variance does not change with treatment? Thus, I am not sure what the point of this is unless the authors are suggesting that a correction factor can be used because the mean observation is lower in the experimental treatment but

the variance does not change? Does the variance change or just the means? The authors show that the values are correlated but what does this really tell us?

This statement does not apply to all analyses. In some analyses in the present manuscript, we found that treatments do not produce strong rank correlations with controls, because the rate at which different samples degrades in response to the treatment is different. In the present analysis, as in many in the present manuscript, the very strong correlation is a Spearman's Rank correlation. This tells us specifically that the rank order of samples, in terms of which samples were high vs low, is basically unchanged. The strong rank correlation shows readers that the effects of the treatment are not such that they are likely to change the nature of study results, as whether samples concentrations are high or low with respect to each other is unaffected by the treatment. This is explained in the Discussion (Lines 697-700).

24. Lines 510, 513, 529, 532, 549, 560: The results are often reported that the rank order of samples between controls and experiments remain "basically" unchanged or "largely" unchanged. I see this vague language as problematic and would rather see a percentage of changes in rank order reported instead of using this language.

Regarding the question of quantifying changes in rank order, in this case, this is precisely what is being assessed, since the correlations used are Spearman's Rank correlations. The language on this has been clarified as such in all places (Lines 514-516; 527-529; 542-546; 546-549; 564-567; 576-578; 588-592).

25. Line 548: same comment as number 23 – with one freezer thaw cycle the values decline but are correlated with controls – suggesting a consistent decline in mean values and similar variance. Can the authors confirm this and what is the biological meaning of reporting this – are the authors suggesting a correction equation could be applied if thawed vs non-previously thawed samples are used in the same analysis?

As the Reviewer states, this is the same comment as number 23, and we have the same response. The very strong rank correlation shows that relative sample concentrations are not strongly affected by the treatment. This is explained in the revised Discussion (Lines 711-712).

26. Line 576: I am still convinced of the applicability of this assay in general to free ranging mammals. I am convinced that this human kit may be applied potentially to come other primate species if the samples have been collected and frozen and not thawed. For macaques and other primates maybe, but I am not convinced this would work for other mammals so this language should be removed.

We have removed the reference here to other mammals (Line 596), and have toned down our language in the Discussion on this point more generally. Nonetheless, we believe that it has promise for other large mammals. We are not concerned about the low recovery because it is so consistent. We also note that the genomic data suggest conservancy of this molecule. Finally, we in any case seek here only to encourage others to investigate. We give substantial text in the Discussion over to advocating that all researchers should undertake their own validations for each study species before utilizing any new biomarker in their studies (Lines 761-785).

27. 584-585: The language used here should be more cautious. I think it is important to note that this method can be used in only certain field conditions where there is reliable electricity for a freezer.

We thank the reviewer for this comment and agree. We have moderated the language at this point and added the caveat regarding freezer access (Lines 683-684).

28. Lines 620-630: I would like a deeper discussion of why suPAR is a superior marker to uNEO and under what conditions one would want to use suPAR instead of uNEO in the field. Given that uNEO is more stable to field conditions, it would seem that it may be advantageous.

Please see our comments above regarding the measurement of infection vs inflammation, and the accompanying changes that we have made to the manuscript.

29. Line 660 and 666: I would like to see this “largely” quantified.

This has been addressed as described in response to the comments above.

30. Lines 672-673: It seems like there was actually an impact – 2 days after there was a significant difference from the controls unless I am reading the data incorrectly?

Yes, there was a statistically significant decline after 2 days of storage in the fridge. This decline was small, however, and the correlation of values on day 2 with those of the controls was very strong ($r=0.98$), indicating that storing at cooling temperature for up to 2 days does not impact the relative differences between sample concentrations. We have clarified this in the revised Discussion (Lines 728-729).

31. Line 683-687: While smaller aliquots may prevent multiple freezer thaw cycles, they also are more likely to thaw completely in transit if dry ice or liquid nitrogen is not used. When samples are aliquoted in 2ml tubes, they are much less likely to thaw compared to those that are aliquoted in say 200ul samples. Thus, there are costs and benefits to each method and this should be noted.

We have added this point to the discussion here (Lines 728-729).

32. Line 703 – except for the very low recovery with macaque urine. What was the recovery with human urine?

Recovery with human urine was not tested in the present study. It would also not be relevant for the present study of macaques, or for other primates.

33. Line 739: I think this is a bit of an overreach given that the human kit revealed very low recovery in a non-human primate species – do the authors really think this kit holds great promise for field studies of large terrestrial mammals (especially given the soil results?).

We believe that the low recovery is not a major issue here, for the reasons we have outlined above, and that given the evolutionary conservancy of the suPAR molecule (see above), it is likely to be similarly measurable in other mammals. We would like to encourage others studying large terrestrial mammals to investigate suPAR measurements. As in all such cases, we strongly recommend new validations for each species (Lines 761-785).